# Crossover interference mediates multiscale patterning along meiotic chromosomes

Martin A. White [1], Beth Weiner[1], Lingluo Chu [1,3], Gyubum Lim [1], Mara Prentiss[2] & Nancy Kleckner [1] ✉

Meiotic crossover interference is a one-dimensional spatial patterning process that produces evenly-spaced crossovers. Quantitative analysis of diagnostic molecules along budding yeast chromosomes reveals that this process sets up two interdigitated patterns, of shorter and longer periodicity, by "two-tiered" patterning. Both tiers comprise clustered assemblies of three types of molecules ("triads") representing the three major components of meiotic chromosomes (crossover recombination, axes, and the synaptonemal complex). One tier of triads occurs at sites of majority ("canonical") crossovers. Second tier triads are more widely spaced but also exhibit interference, dependent on the same functions as canonical crossover interference. Diverse lines of evidence suggest that second tier triads arise at sites of previously mysterious "minority" crossovers. Finally, conserved protein remodeler Pch2/TRIP13 modulates the abundance of triad components, specifically in longer periodicity triads, dynamically in real time. Potential roles of triad structure, mechanisms of two-tiered patterning, and the nature of minority crossovers are discussed.

Meiosis is the cellular program that produces haploid gametes from diploid progenitor cells as required for sexual reproduction. A core component of meiosis is the shuffling of alleles between the maternal and paternal versions of each chromosome ("homologs"), by crossover recombination[1]. As discovered over a century ago by genetic analysis in *Drosophila melanogaster*, crossovers occur at different positions in different nuclei, with occurrence of a crossover at one position "interfering" with occurrence of another crossover nearby. This communication between neighboring crossovers implies that crossover formation is regulated by a one-dimensional patterning process. The ultimate effect of this process is that, along any given chromosome, crossovers tend to be evenly spaced[2,3]. This outcome is evolutionarily important because even spacing, per se, enhances crossover-mediated allelic shuffling[4]. It is also important for fertility because crossovers have a mechanical role to ensure regular segregation of homologs to opposite poles at the first meiotic division, and regular spacing is important for this process[5]. Crossover interference

thus provides an interesting example of one-dimensional spatial patterning, with significant evolutionary and functional implications, in an important biological system.

Crossover positions can be defined not only genetically, but also by localization of cytologically-visible, crossover-specific, protein/DNA recombination complexes. This was first appreciated when electron microscopy revealed prominent "recombination nodules" that are arrayed with an interference distribution along the synaptonemal complex (SC); a conserved structure that links homolog axes along their lengths at mid-prophase (pachytene)[6] (Fig. 1A, B). These crossover complexes are also detectable as SC-associated foci of appropriately labeled components by fluorescence microscopy (Fig. 1C). In some cases, they have also been examined in detail by super-resolution fluorescence microscopy (e.g., ref. 7). Cytological methods for analysis of crossover patterns have a conceptual advantage over genetic or genomic approaches because the metric for crossover interference is physical distance along prophase chromosomes, in micrometers (μm)

[1]Department of Molecular and Cellular Biology, Harvard University, Cambridge, MA, USA. [2]Department of Physics, Harvard University, Cambridge, MA, USA. [3]Present address: Bioscience and Biomedical Engineering Thrust, Systems Hub, Hong Kong University of Science and Technology (Guangzhou), Guangzhou, People's Republic of China. ✉e-mail: kleckner@fas.harvard.edu

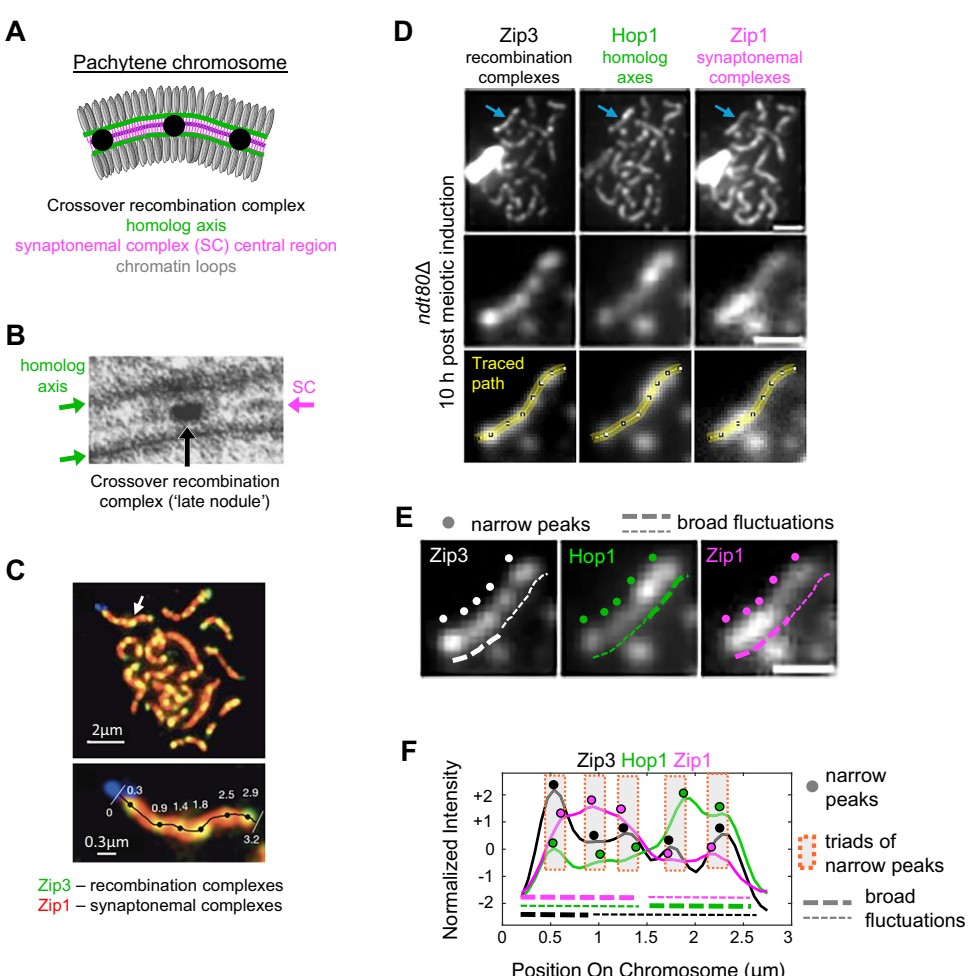

**Fig. 1 | Meiotic Chromosomes. A** Cartoon of pachytene chromosome structure, with key components indicated. **B** micrograph of *Sordaria macrospora* meiotic pachytene chromosome acquired by electron microscopy (adapted from ref. 57 under a CC BY-NC 4.0 license). (**C**, adapted from ref. 8) Top: Zip3 foci (green) along pachytene synaptonemal complexes of budding yeast (marked by Zip1 protein; red). *Bottom:* positions of Zip3 foci which define canonical crossovers. **D** Top, micrographs of spread chromosomes from a pachytene arrested (*ndt80Δ*) cell of budding yeast that have been immunostained, simultaneously, for Zip3, Hop1 and Zip1 proteins. Arrows indicate example chromosome (representative from total sample size of 81 chromosomes). Scale bar, 2 μm. *Middle*, enlarged image of example chromosome. Scale bar, 1 μm. *Bottom*, path (yellow) traced through the

mid-line of example chromosome to extract its signal intensity profiles. **E** Narrow and broad signal features suggested by quantitative signal intensity profiles (below, **F**). Narrow signals of Zip3 correspond to Zip3 'foci' that define canonical crossovers (e.g., **C**). **F** Normalized signal intensity profiles of Zip3, Hop1 and Zip1 for the example chromosome in (**D**) suggest broad signal fluctuations punctuated by narrow peaks, for all three molecules. Narrow peaks of the three molecules colocalize in "triads". Further work (below) indicates that this is also true for peaks of broad signal fluctuations. Normalized intensity is defined, for each signal, as the number of standard deviations from the mean. Source data are provided as a Source Data file.

rather than genetic (centiMorgans) or DNA (Megabase pair) distance[8]. Thus, the positions of crossover complexes along organized chromosomes (e.g., Fig. 1C bottom) directly report the outcome of the crossover patterning process.

It is generally thought that a single patterning process gives rise to these cytologically-detectable interfering crossovers (hereafter "canonical" crossovers). However, in several organisms, including budding yeast, the number of crossovers detected genetically, or by DNA sequence analysis, is significantly greater than the number of cytologically-defined complexes[9]. This disparity points to the existence of a second, minority set of crossovers. The nature of these minority crossovers has been mysterious (below).

At the mechanistic level, crossover patterning is imposed upon a large array of so-called "precursor" recombination interactions. These interactions comprise nascent interhomolog DNA linkages that span paired homolog structural axes all along the chromosomes[10]. A small subset of these precursors is designated for eventual maturation into canonical crossovers, by the rules of

spatial patterning, while the remaining interactions are matured to other fates. Current models propose that minority crossovers are aberrant outcomes of non-crossover fated interactions (e.g., ref. 11.).

Emergence of any spatial pattern requires communication within the system. All current models suggest that the communication required for patterning of crossovers involves transmission of a signal along longitudinal structures, either the axes of individual homologs and/or the SC[10]. In budding yeast, crossover interference is imposed prior to SC formation, with communication occurring along axes, dependent on axis components[8,12]. Communication along these structures can confer the requisite effects on DNA events of recombination because, universally, biochemical recombination complexes are physically and functionally associated with underlying structural elements at all stages[10].

In the present study, we further explored the localization patterns of, and relationships among, crossover-related components and axis and SC components in the budding yeast *Saccharomyces cerevisiae*.

In budding yeast (and analogously in other organisms) formation of canonical crossovers is mediated by a set of physically and functionally interacting molecules known as ZMMs[1,10,13]. These ZMMs include: Msh4/5 (also called MutSγ), which binds and stabilizes branched DNA structures[7,14]; Zip3, an E3 ligase[15]; and the ZZS complex (Zip2, Zip4, Spo16), which mediates the recombination complex/axis-SC interface[16,17]. Canonical crossover positions have been defined, and their patterns analyzed, by the disposition of fluorescent foci of ZMMs Zip3 and Zip2 along SCs[8,9] (e.g., Fig. 1C), which colocalize along pachytene chromosomes[15] (Supplementary Fig. 1). Additional studies show that, in budding yeast, the early precursor recombination complexes are themselves evenly spaced with a modal distance of 0.23 μm[18]. This feature has the important implication that inter-crossover distances will be quantized in units of inter-precursor distance (below).

As seen by electron microscopy, the SC is a highly regular structure with defined edges (the lateral elements) linked by close-packed transverse filaments that emanate from a prominent midline central element[19] (Fig. 1A, B). Chromatin loops emanate outward to either side of the structure (Fig. 1A, B). Each lateral element marks the inner layer of a more complex axial meshwork that includes meiosis-specific components, notably a conserved filamentous component (Red1 in yeast) and its partner, conserved HORMAD protein(s) (Hop1 in yeast), plus components also found along mitotic[19,20] chromosome axes. Molecules of the SC transverse filament protein (Zip1 in yeast) have their C-termini in contact with the axes/lateral elements and their N-termini in contact with the central element[19].

Although electron microscopy images of the SC define a highly regular structure (above), florescence microscopy images of budding yeast axis and transverse filament proteins, Hop1 and Zip1, reveals that both molecules are distributed non-uniformly along chromosomes[21]. Both tend to occur as broad (-1–1.5 μm) regions of higher or lower signal intensities (a measure of protein abundance), often with different relative intensities for the two molecules (Fig. 1D middle and right columns). Regions of high signal intensity are sometimes referred to as "hyperabundance domains". The observed fluctuations in signal intensities are actively created by the conserved protein remodeler, AAA + -ATPase Pch2 (also known as TRIP13): in the absence of Pch2, fluorescence microscopy images of both molecules exhibit a higher intensity, smoother signal along chromosomes (refs. 21,22; below). Pch2-mediated modulation of chromosome binding involves complex interactions among Hop1 and Zip1 (below). The significance of these broad domainal fluctuations is unknown.

As a further complication to the localization patterns of crossover-related components and axis and SC components, the distinction between focal and domainal localization is not absolute. Axis component Hop1 exhibits not only broad abundance domains, but also brighter focal densities in fluorescence microscopy images that seem to correlate with foci of Zip3[23]. Also, SC component Zip1 occurs not only within the SC but also has local roles for recombination that precede its installation in the SC, as well as being required for normal Zip3 association with other ZMMs[24,25]. Conversely, by genomic analysis, crossover protein Zip3 is not only directly associated with recombination complexes in focal signals, but also localizes to chromosome axis association sites (thus, axes)[26,27] and, as shown below, also loads at a low level with a domainal character.

## Results
### Experimental approach
The present study analyzed the localization patterns of Zip3 (or Zip2), Hop1, and Zip1 proteins in widefield fluorescence microscopy images of individual pachytene chromosomes of budding yeast that had been immunostained for all three molecules (Fig. 1D top and middle rows). Previous yeast cytological studies have defined localization patterns for these molecules by analysis of conveniently thresholded images

(e.g., ref. 8; Fig. 1C bottom). Here, instead, we defined and analyzed quantitative signal intensity profiles of the three molecules, first in wild type meiosis and then in mutants known to exhibit reduced crossover interference and/or lacking Pch2. Signal intensity profiles reflect the distribution of protein abundance along each measured chromosome.

Signal intensity profiles of Zip3, Hop1 and Zip1 were generated for 81 wild type chromosomes that had been extracted from the nuclei of cells arrested in pachytene (using an *ndt80Δ* mutation). These intensity profiles were defined by tracing a path along the length of each chromosome in Fiji (Fig. 1D bottom) and extracting the pixel intensity values along that path for each of the three molecular components, using the Fiji Plot Profile function (Methods; Fig. 1F and Supplementary Fig. 2). Specific chromosomes were not identified. However, chromosomes selected for tracing were narrowly distributed in length around 3.15 ± 0.4 μm (mean ± standard deviation; Supplementary Fig. 3), very similarly to chromosome XV, the second largest chromosome in yeast, comprising 1.1 Mb (3.14 ± 0.3 μm; Supplementary Fig. 3). Critically for our analysis, bleed-through of each signal into the acquisition channels for the other two signals is negligible (Supplementary Fig. 4).

A parallel experiment analogously examined wild type chromosomes stained for Zip2, Hop1 and Zip1 ($n = 56$ chromosomes). Indistinguishable results were obtained in the two cases (below). Notably, this experiment used (i) pachytene chromosomes of a non-arrested culture, implying that pachytene arrest does not alter the results, and (ii) a different combination of fluorescently-labeled secondary antibodies, implying that the results do not depend on the specific image excitation and acquisition wavelengths (Methods).

Inspection of individual signal intensity profiles for the three types of molecules alerted us to two features of interest. First, all three molecules seem to exhibit both narrow and broad signals, with narrow peaks superimposed on broader fluctuations (Fig. 1E, F dots and lines, respectively). Interestingly, the different localization patterns reported previously for the three different molecules (above) can be explained by the relative prominences of the two types of signals in the three cases: narrow signals are more prominent for Zip3 (i.e., as "foci", Fig. 1C) for which broader signal fluctuations have not been previously reported; Hop1 exhibits an intermediate pattern, in accord with previous work[23]; and broader ("domainal") fluctuations are more prominent for Zip1, for which narrow/"focal" signals have not been previously been discerned. Second, for narrower signals, intensity peaks of the three molecules appear to colocalize with one another, in "triads" (Fig. 1F boxes).

### FFT analysis defines shorter and longer periodicity signal intensity fluctuations which are the same for all analyzed molecules
To further understand the qualitative features suggested by visual inspection, we used Fast Fourier Transform (FFT) analysis of the data to identify whether prominent fluctuations in signal intensity are characterized by specific spatial periodicities. A Fourier transform describes any pattern of interest as the sum of a set of sine and cosine waves of varying amplitude and frequency. The Fourier transform of each individual signal intensity profile (e.g., Fig. 2A) is a distribution of the amplitudes of corresponding peaks in k-space, where k is the Fourier transform vector (e.g., Fig. 2B). This distribution can then be converted to position-space (μm; Methods).

Given the visual impression of two general types of periodicities above (Fig. 1D–F), we first analyzed FFT outputs by defining the two most prominent periodicities for each transform, given by the two peaks (local maxima) of highest amplitude in the frequency range of interest (e.g., Fig. 2B asterisks). The distributions of these most prominent periodicities across all chromosomes were then plotted, for each of the three molecules. All three distributions can be described by the sum of two broad Gaussian distributions, centered on -0.45 μm and -1 μm for all three molecules (Fig. 2C). Similar results are seen for

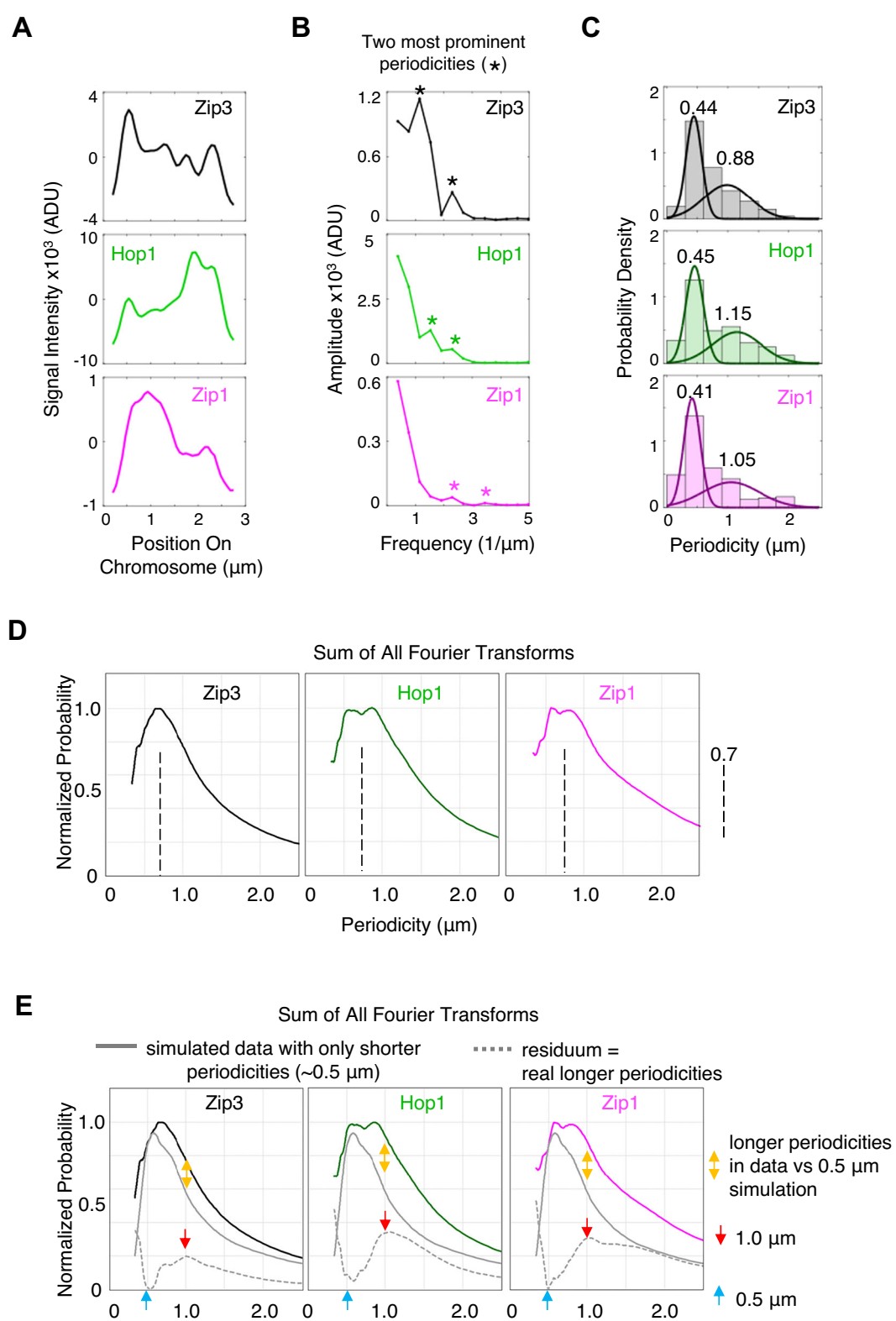

Zip2-Hop1-Zip1 analysis (Supplementary Fig. 5A). These two periodicities correspond well to those seen by visual inspection. We also note that the shorter periodicity is the known distance between adjacent canonical crossovers as defined by Zip3 foci in standard analyses (~0.5 μm; below).

These distributions of most prominent peaks are potentially subject to an artifact. If one takes the Fourier transform of a finite shape (e.g., a chromosome), k-vector periodicities can arise simply as a function of confinement of the signal to that shape, rather than reflecting true periodicities in the analyzed data (e.g., Supplementary Fig. 6A–G). We therefore carried out a second type of analysis that controls for this and other potential artifacts (below; Supplementary Fig. 6).

First, for each molecule, the individual Fourier transforms for all chromosomes were summed. The resulting periodicity probability

**Fig. 2 | Fourier transform analysis defines two groups of signal intensity fluctuations, of shorter and longer periodicities. A** Signal intensity profiles for the three molecules of interest along example chromosome of Fig. 1E, F; ADU, analog-to-digital units. **B** FFT amplitude spectra from Fourier analysis of signal intensity profiles in (**A**), plotted in k-space. Asterisks indicate the two highest amplitude peaks. **C** Distributions of the two highest amplitude peaks in the FFT amplitude spectra across all measured chromosomes ($n = 81$ chromosomes). Best fit two-component Gaussian mixture models (means indicated) are consistent with two groups, of shorter and longer periodicities, for all three molecules. See also Supplementary Fig. 5A. **D, E** The sums of Fourier transform periodicities for all chromosomes can be deconvolved into shorter and longer periodicity groups centered at -0.5 μm and -1 μm. **D** For each molecule, all individual Fourier transforms were summed in k-vector space and the sum converted to a distribution of probabilities in position space (μm; Methods). **E** Solid gray line: the distribution of total periodicity probabilities was determined for simulated chromosome data that matches the shorter periodicity (-0.5 μm) peaks visible in primary experimental data and known for Zip3 foci (Supplementary Fig. 6; Fig. 4D). When this distribution is subtracted from the experimental distribution of total periodicities (**D**), the residuum (dashed gray line) exhibits no periodicities at -0.5 μm (blue arrow). Thus, shorter periodicities in the data are fully explained by the subtracted shorter periodicities. A significant residuum of longer periodicities occurs (dashed gray line) centered at -1 μm (red arrow), over and above longer periodicities present in simulated data (gold arrows), implying that this set of longer periodicities is present in experimental data (see also Supplementary Fig. 6). Source data are provided as a Source Data file.

distributions were converted from k-vector space to position-space. The Zip1 and Hop1 signals are both seen to exhibit two peaks, implying the presence of two periodicities of greater and lesser than -0.7 μm (Fig. 2D; dashed vertical lines). This outcome is in accord with the two groups observed in the distributions of the two most prominent periodicity peaks of each transform (-0.5 μm and -1 μm; above). Zip3 exhibits a broad peak centered between these two periodicities (Fig. 2D). This different outcome can be explained by the fact that, for Zip3, the shorter periodicity signal is much more prominent, relative to the longer periodicity signal, than for the other two molecules (Background; below).

Further analysis of these probability distributions reveals that they can be explained as the sum of two groups of shorter and longer periodicities, each characterized by substantial heterogeneity.

- We first simulated a set of signal intensity profiles that matched the -0.5 μm peaks visible in primary experimental signal intensity profiles (e.g., dots in Fig. 1E, F; Supplementary Fig. 6H). Each simulated profile comprised a series of Gaussian distributions with an average inter-peak distance of -0.5 μm. Variations included significant heterogeneity in inter-peak distances corresponding to those defined for experimental signal intensity profiles and, for Zip3, inter-focus distances in standard analyses. Peak heights and chromosome lengths were both also varied appropriately. Inclusion of inter-peak distance heterogeneity in the simulated data is especially important (Supplementary Fig. 6H, I versus Supplementary Fig. 6C–E).

- We then determined the Fourier transforms for -100 such simulated signal intensity profiles, summed those transforms in k-vector space, converted the sum to position-space, and subtracted that sum from the total probability distribution defined experimentally. If the distributions of total summed periodicities (Fig. 2D) are made up entirely of broad shorter and longer periodicity distributions of the type described above (Fig. 2C), subtraction of the simulated sum for only the shorter periodicities should cause the probability distribution to fall to near zero at -0.5 μm, after which it should rise in a broad peak centered at -1 μm, corresponding to the heterogeneous set of longer periodicities. Exactly these features are observed (Fig. 2E, dashed gray line indicates residuum). We also note that the residual longer periodicity peak for Zip3 is lower than those for Hop1 and Zip1, in accord with the fact that broad signals are less prominent, relative to narrow signals for this molecule (Background). This difference is also discernible in the two-most-prominent-peaks distribution (Fig. 2C). We also note that the outcome of this subtraction analysis is quite robust with respect to the exact variation parameters chosen for simulations of the shorter periodicity group.

In summary, the above analysis shows that, for each of the analyzed molecules, the total distribution of periodicity probabilities can be deconvolved into two groups, shorter and longer, of -0.5 μm and -1 μm respectively, with significant heterogeneity in inter-event spacings in both cases.

We also note that the above-identified periodicities are well above the resolution limit of our imaging system (190–240 nm; Abbe's diffraction resolution limit of light).

## Inverse FFT analysis reveals that Zip3 (Zip2), Hop1 and Zip1 occur in shorter and longer periodicity triads

To analyze the shorter and longer periodicity signals in greater detail, we applied inverse FFT ("iFFT"). This method defines, for an individual analyzed sample, the fluctuations in signal intensity that are given by any selected subset of FFT-defined periodicities. In the present case, we divided the total periodicities into two groups of less than, or greater than, 875 nm. iFFT outputs for each molecule, of each individual chromosome, defines the fluctuations in signal intensity corresponding to shorter and longer periodicity signal components for that case.

The outcome of such analysis is shown in Fig. 3A for the example chromosome of Fig. 1D–F. For all three molecules, iFFT of the shorter periodicity group gives rise to a regular series of narrow closely spaced peaks (Fig. 3A middle column) while the longer periodicity group gives rise to a regular series of broader, widely spaced peaks whose signal intensities vary along the length of the chromosome, differently for different molecules (Fig. 3A right column). These patterns correspond to the visually identified narrower and broader signals discussed above. Moreover, peaks of the three molecules occur in triads, for both periodicity groups (Fig. 3B).

For detailed quantitative analysis, the position of each peak in each iFFT profile was defined at one pixel resolution (67 nm). Relationships among different sets of peaks were then examined across all chromosomes as desired. We note that defining peaks in this manner allows the distances between peaks of different types (either shorter/longer periodicity iFFT peaks of different molecules on the same chromosome, or shorter versus longer periodicity iFFT peaks of the same molecule on the same chromosome), to be determined below the resolution limit of imaging, as illustrated below.

Analysis of inter-peak distances along each iFFT-derived signal intensity profile confirms that all three analyzed molecules exhibit the same two major periodicity groups. The distributions of distances between adjacent peaks of the same molecule exhibit modal values 0.49–0.51 μm and 0.93–0.98 μm for shorter and longer periodicity iFFT peaks respectively (Fig. 3C, D), matching the -0.5 μm and -1 μm periodicity groups defined by Fourier analysis above.

Further analysis demonstrates that the intensity peaks for each of the three molecules are clustered in "triads", for both periodicity groups, across all chromosomes. In a triad pattern, a peak of any given molecule and the nearest peak of a different molecule should be very close together; and these distances should be much closer than expected if the positions of the two peaks were determined independently. Accordingly, the modal distances between nearest heterologous peaks are 0.07 μm (67 nm) in all cases, much less than expected for independence (0.27–0.28 μm and 0.48–0.50 μm for shorter and longer periodicity iFFT peaks respectively; Fig. 3E, F dashed lines). Additionally, the distance between adjacent triads should be large as compared to intra-triad spacings. If so, the distance between a given peak and the nearest adjacent heterologous peak (which should usually be within the same triad) should be much smaller than the

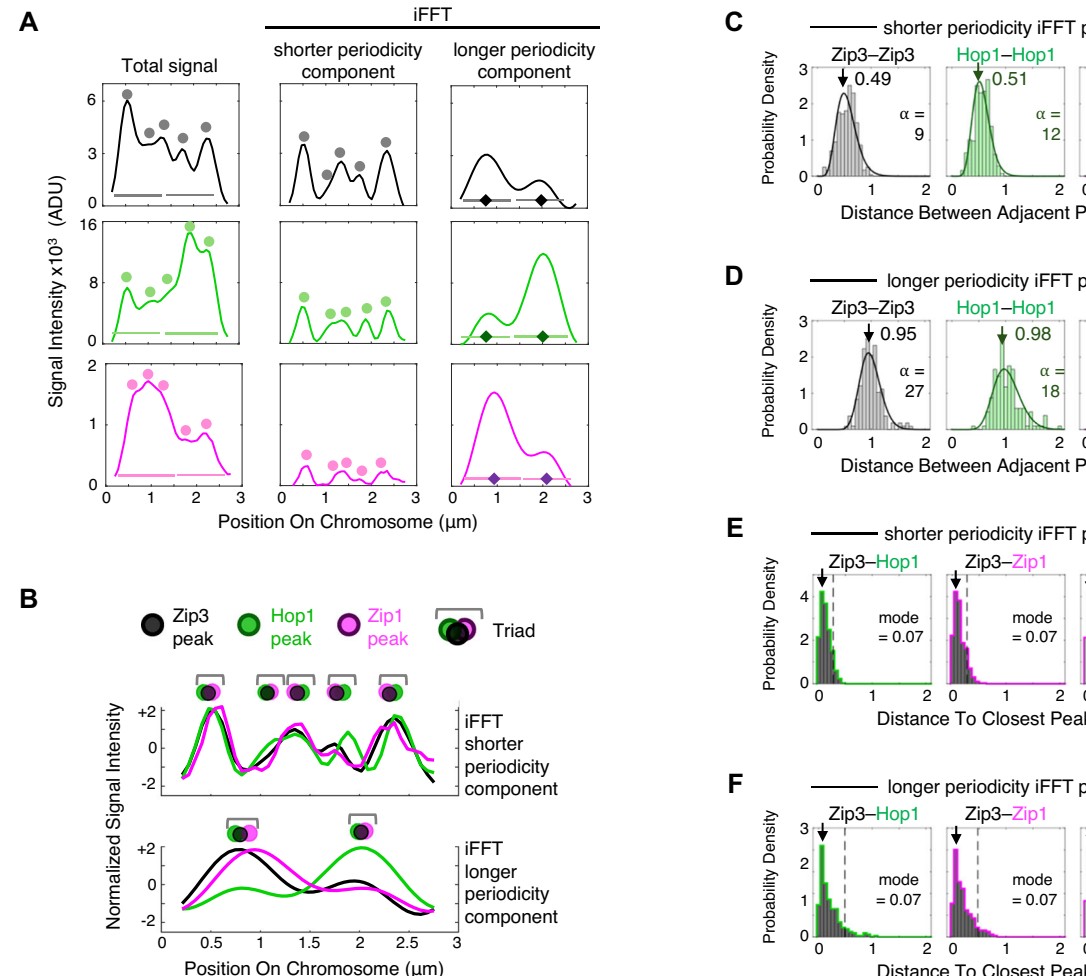

**Fig. 3 | Fourier transform analysis defines two groups of signal intensity fluctuations, of shorter and longer periodicities, whose peaks occur in Zip3/Hop1/Zip1 triads. A, B** Inverse FFT (iFFT) was applied to the two groups of periodicities of less than, and greater than, 875 nm. Shorter and longer periodicity signal components for each molecule were reconstructed for each individual analyzed chromosome (e.g., example chromosome of Fig. 2A). **A** Shorter and longer periodicities give rise to narrow and broad peaks respectively, similar to the narrow and broad fluctuations suggested by visual inspection (compare with Fig. 1E, F). **B** Overlays of iFFT outputs for the three molecules illustrate clustering of their peaks in triads, for shorter and longer periodicity signal components. Distances between adjacent

iFFT-defined peaks match the input periodicities for shorter (**C**) and longer (**D**) periodicity signal components (compare with Fig. 2C). Best fit gamma distributions (solid lines) with modes (arrows) and shape parameters (α) are shown. See also Supplementary Fig. 5B, C. **E, F** Triad structure occurs analogously for both groups of periodicities across all chromosomes. For both groups, distances between each peak and the nearest peak of a different molecule are very small (modal distance = -0.07 μm), much smaller than expected by chance (-0.3 μm and -0.5 μm; dashed lines), for all three pairs of molecules; see also Supplementary Fig. 5D, E. Source data are provided as a Source Data file.

distance between that given peak and the nearest signal of that same molecule (which should be in the adjacent triad). Correspondingly, the 0.07 μm spacings between heterologous peaks are much smaller than the distances between adjacent homologous peaks (-0.5 μm and -1 μm; Fig. 3C, D; above).

Comparable results were obtained for iFFT signal profiles of Zip2, Hop1 and Zip1 along wild type chromosomes not subject to pachytene arrest (Supplementary Fig. 5B–E).

**Shorter periodicity triads arise at sites of canonical crossovers**
The existence, positions and patterns of canonical crossovers have been defined previously by standard analysis of Zip3 and Zip2 foci[8,9,18] (Background; Fig. 1C). Shorter periodicity iFFT peaks of either molecule exhibit patterns that are essentially identical to those of canonical crossovers, in every basic characteristic respect. The same is true for shorter periodicity iFFT peaks of Hop1 and Zip1, in accord with triad structure.

Canonical crossovers are characterized by crossover interference, which is classically defined by coefficient of coincidence (CoC)

analysis[2,3] (Fig. 4A). In brief, a chromosome is divided into intervals; for each pair of intervals, the observed number of chromosomes with a crossover in both intervals is divided by the number expected if events occurred in the two intervals independently. This ratio (the CoC) is then plotted as a function of inter-interval distance[28]. For canonical crossovers, this ratio is very low (often zero) at small inter-interval distances, because designation of a crossover at one position strongly "interferes" with designation of another crossover nearby. This ratio increases progressively until interference is absence (CoC = 1), with the distance required to reach this level being lesser or greater according to the (smaller or larger) distance over which the interference inhibition acts.

A useful metric is $L_{CoC}$, the inter-interval distance at which CoC = 0.5. For canonical Zip3-defined crossovers, $L_{CoC}$ = 0.27 and 0.31 for chromosomes XV and III, respectively[8] (Fig. 4B). $L_{CoC}$ values for shorter periodicity iFFT peaks are the same, not only for Zip3 and Zip2, but also for Hop1 and Zip1 ($L_{CoC}$ = 0.27–0.32 μm; Fig. 4B, C top, and Supplementary Fig. 7A).

Meiosis-specific depletion of Topoisomerase II has previously been shown to reduce interference among canonical crossovers

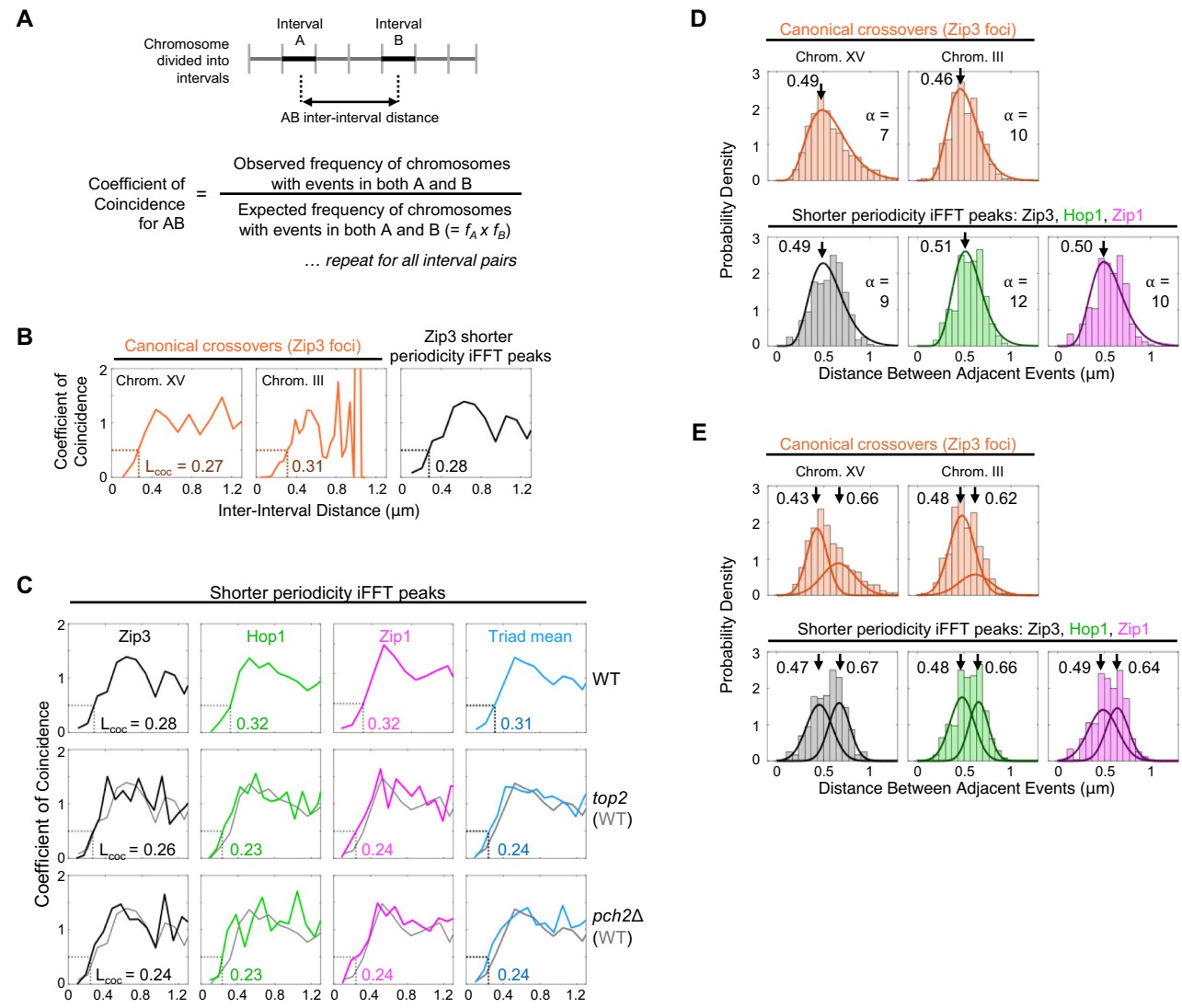

**Fig. 4 | Focal triads arise at sites of "canonical" crossovers.** Patterns of canonical crossovers, as defined by standard Zip3 focus analysis, along chromosomes III (n = 683, data from ref. 8) and XV (n = 206, this work) are compared to the patterns of shorter periodicity iFFT peaks (n = 81 chromosomes unless otherwise stated). See also Supplementary Figs. 7–9. **A** Method for calculating the coefficient of coincidence, a metric for quantifying communication between crossovers on the same chromosome (see also text). For **B**, **C** chromosomes were divided into 30 intervals; $L_{coc}$ is the inter-interval distance at which CoC = 0.5. **B** *Left*, CoC curves for canonical crossovers on chromosomes III and XV. *Right*, CoC curve for Zip3 shorter periodicity iFFT peaks. **C** CoC curves for shorter periodicity iFFT peaks of Zip3, Hop1 and Zip1 proteins, and the mean CoC curve for all three triad

components ('Triad mean'), on chromosomes isolated from either a wild type strain ('WT', top; n = 81), a strain depleted for Topoisomerase 2 ('*top2*', middle; n = 60), or a strain lacking Pch2 ('*pch2Δ*', bottom; n = 52 chromosomes). Top left panel is a reproduction of (**B**) right. **D** distributions of distances between adjacent canonical crossovers (*top*) and adjacent shorter periodicity iFFT peaks of Zip3, Hop1 and Zip1 (*bottom*; reproduction of Fig. 3C) are the same with respect to modes (arrows) and shape parameters (α) of best-fit gamma distributions (solid lines). **E** Distributions of distances between adjacent events of all types (from (**D**)) are well-fit by two Gaussian distributions, with the same mean values for canonical crossovers (*top*) and shorter periodicity triad peaks (*bottom*). Source data are provided as a Source Data file.

(as defined by Zip3 foci). This effect is manifested as a shift of CoC curves to the left and a corresponding reduction in $L_{CoC}$ (from 0.27 in wild type to 0.24 in the mutant[8]; Supplementary Fig. 7B orange lines). Shorter periodicity iFFT peaks are reduced by Topoisomerase II depletion to the same extent seen for canonical crossovers, for each of the three triad components ($L_{CoC}$ = 0.28 – 0.32 μm in wild type versus 0.23 – 0.26 μm in the mutant (n = 60 chromosomes; Fig. 4C middle and Supplementary Fig. 7B black lines).

Crossover interference is also reduced in the absence of AAA + -ATPase Pch2, as defined by genetic analysis[23,29]. A *pch2Δ* mutation also reduces interference among shorter periodicity iFFT peaks, to the same level seen upon Topoisomerase II depletion, not only for Zip3 but also for Hop1 and Zip1 ($L_{CoC}$ = 0.23 μm – 0.24 μm; n = 52 chromosomes;

Fig. 4C bottom). Importantly, neither mutation affects triad formation (Supplementary Fig. 8).

Interference among canonical crossovers, as defined by CoC analysis, is also reflected in the distribution of distances between adjacent crossovers. These distributions are often described by the properties of their best-fit gamma distributions (e.g., refs. 18,30.). For canonical crossovers (as defined by Zip3 foci) along chromosomes XV and III, the best-fit gamma distributions have modal distances of 0.49 μm and 0.46 μm, and shape parameters values of 7 and 10, reflecting relatively even spacing (Fig. 4D top). The gamma distributions that best describe the distances between adjacent shorter periodicity iFFT peaks of Zip2, Zip3, Hop1, and Zip1 are indistinguishable from those for canonical crossovers, with modal distances of

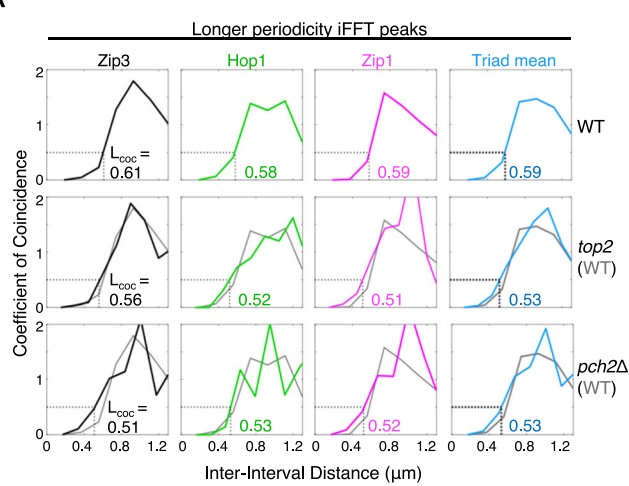

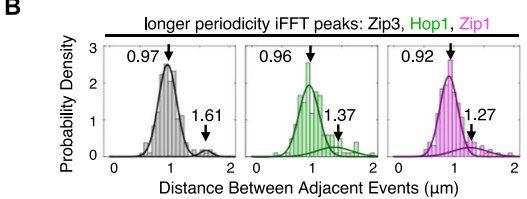

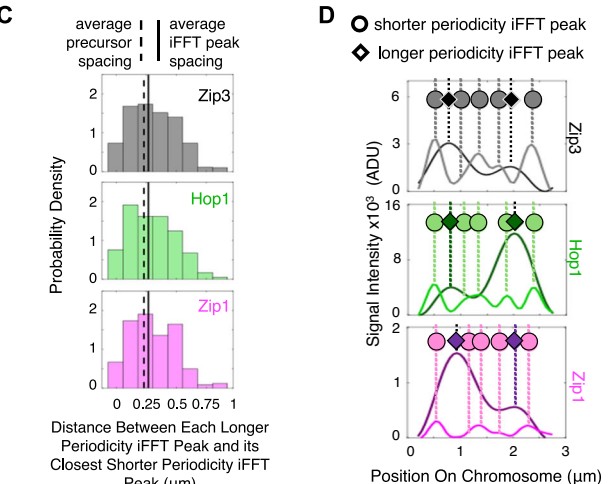

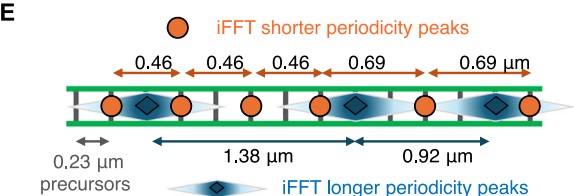

**Fig. 5 | Longer periodicity triad peaks and their relationship to shorter periodicity triad peaks (i.e., canonical crossovers). A** CoC analysis shows that longer periodicity iFFT peaks exhibit interference. This interference is stronger than the interference observed for shorter periodicity iFFT peaks/canonical crossovers (*top*, compare $L_{CoC}$ values with Fig. 4B, C *top*); but it is analogously reduced in a *top2* depletion mutant (*middle*) and *pch2Δ* strain (*bottom*) compared to wild type (compare with Fig. 4C *middle, bottom*). Data are from the same experiments as in Fig. 4C. See also Supplementary Fig. 10A. **B** Distributions of distances between adjacent longer periodicity iFFT peaks, of all three molecules, are well described by two Gaussian distributions (solid lines) with the same mean values in all cases, pointing to two subcategories of spacings. (**C**) The median distance between each longer periodicity iFFT peak and its nearest shorter periodicity iFFT peak (solid vertical lines; $n = 81$) matches the expected distance between adjacent early recombination (pre-crossover) precursors (0.23 μm; dashed lines; text), suggesting that both types of triads arise along this same basic array of early recombination precursor sites. See also Supplementary Fig. 10B. **D** Interdigitation of shorter and longer periodicity iFFT peaks illustrated for the example chromosome of Fig. 3A, B. **E** Illustration of interdigitated shorter and longer periodicity triads, with the two different sub-periodicities observed in each case, along a common array of precursor sites evenly spaced at ~0.23 μm. Shorter periodicity triad peaks correspond to sites of canonical crossovers. Longer periodicity triad peaks are proposed to correspond to sites of minority crossovers (text; Fig. 6). Source data are provided as a Source Data file.

acts (so-called "precursors") are evenly spaced at an average distance of ~0.23 μm (Background). As a result, inter-crossover distances will be quantized in units of the inter-precursor distance. The two inter-focus separations defined by the two Gaussian distributions correspond closely to two (~0.46 μm) and three (~0.69 μm) times that inter-precursor distance. That is, crossovers are usually separated by two inter-precursor distances but, less frequently, are separated by three inter-precursor distances.

The distributions of distances between adjacent shorter periodicity iFFT peaks exhibit these same features. Distributions for Zip2, Zip3, Hop1 and Zip1 are also well fit by pairs of Gaussian distributions with mean values of 0.41–0.49 μm and 0.62–0.67 μm (Fig. 4E bottom and Supplementary Fig. 9). Thus, shorter periodicity triads, which include peaks of Hop1 and Zip1 as well as of Zip3 and Zip2, exhibit quantized spacing similar to that of canonical crossovers.

Taken together, the above correspondences demonstrate that triads of the three types of molecules arise at the positions of canonical crossovers. These correspondences concomitantly provide strong confirmation of the validity of FFT-defined periodicities and the accuracy of iFFT-defined peak positions.

### Longer periodicity triads are patterned by the same interference process as canonical crossover triads

The patterns of longer periodicity triads exhibit strong analogies to those of shorter periodicity triads (and thus canonical crossovers).

Longer periodicity iFFT peaks also exhibit interference as defined by CoC analysis. All three molecules exhibit the same interference, with $L_{CoC}$ of 0.58–0.62 μm (Fig. 5A top and Supplementary Fig. 10A). The strength of this interference is greater than that of shorter periodicity iFFT peaks/canonical crossovers ($L_{CoC}$ of 0.27–0.32 μm, above). This is expected for a regular signal of longer (versus shorter) periodicity.

Interference among longer periodicity iFFT peaks is reduced by the same mutations that reduce canonical crossover interference. For all three molecules, meiotic depletion of Topoisomerase II or deletion of Pch2 reduces $L_{CoC}$ from the wild type value of ~0.59 μm (above) to 0.51–0.56 μm (Fig. 5A middle, bottom vs 5A top). Thus: interference-mediated patterning of longer periodicity triads requires the same functions as interference-mediated patterning of canonical crossovers/shorter periodicity triads. The important general implication of these results is that the canonical crossover interference process sets

0.48–0.51 μm and shape parameters of 9–12 (Fig. 4D bottom and Supplementary Fig. 5B).

Upon further inspection, the total distribution of distances between adjacent canonical crossovers (Zip3 foci) are well explained as the sum of a pair of Gaussian distributions: a more prominent one centered on 0.43 – 0.48 μm and a less prominent one of 0.62–0.66 μm (Fig. 4E top). The presence and relative heights of these two distributions explain why the total distribution is not perfectly symmetric (Fig. 4D top). The early recombination interactions upon which canonical crossover interference

up not one, but two tiers of evenly spaced triads, of shorter and longer periodicities.

The distributions of distances between adjacent longer periodicity iFFT peaks, for all three molecules, are asymmetric (Fig. 3D). These distributions well explained by two Gaussians, whose mean are 0.92–0.97 μm and 1.3–1.6 μm, respectively (Fig. 5B). These two periodicities correspond well to four and six times the 0.23 μm distance between early recombination precursors from which canonical crossovers arise (4 × 0.23 = 0.92 μm and 6 × 0.23 = 1.38 μm respectively). These three properties mirror those seen for shorter periodicity triads/canonical crossovers which also exhibit asymmetric distributions, well-explained by two Gaussian distributions, whose mean values correspond to integral numbers of inter-precursor distances (above). These findings are consistent with the possibility that longer periodicity triads arise along the same set of precursor recombination interactions as shorter periodicity triads but are more widely spaced.

It was also of interest to investigate whether the two types of triads arise at the same, or different, positions along a chromosome and, in the latter case, to define the separation distances between triads of the two types. Since longer periodicity triads are less frequent than shorter periodicity triads, we defined the distance between each longer periodicity iFFT peak of a given molecule and the nearest shorter periodicity iFFT peak of the same molecule, for all three molecules. Two findings emerge.

First, the two types of triads almost always occur at different positions. 82% of all longer periodicity iFFT peaks are separated from a shorter periodicity iFFT peak by at least 100 nm ($n = 1422$ total peak pairs for Zip3, Hop1 and Zip1; Fig. 5C; see also Supplementary Fig. 10B for analogous experiment assaying Zip2, Hop1 and Zip1 in the absence of pachytene arrest). Put another way, shorter and longer periodicity triads are interdigitated along each chromosome, as illustrated for the example chromosome (Fig. 5D).

Second, for all three molecules, the distances between each longer periodicity iFFT peak and its nearest shorter periodicity iFFT peak have identical median spacings of 0.27 μm (Fig. 5C). This corresponds closely to the average distance between two precursor sites (0.23 μm; above), providing further evidence that longer periodicity triads arise along the same set of early recombination precursors as shorter periodicity triads. It also suggests specifically that a longer periodicity triad tends to occur at a precursor site that is immediately adjacent to a precursor site at which a canonical crossover/shorter periodicity triad occurs.

Finally, as described above, individual pairs of heterologous triad component peaks are separated by ~70 nm for longer periodicity triads just as for shorter periodicity triads. Thus, in both cases, the peaks of three components of each individual triad will be clustered within the same ≤~200 nm. Given the analogies between shorter and longer periodicity triads as described above, similar tight clustering of the three molecules within each triad suggests that both types of triads are analogously nucleated at recombination precursor sites. It would therefore appear that the primary difference between the two types of triads, other than their periodicities, is that in the case of shorter periodicity triads/canonical crossovers, spreading from the nucleation site is confined to a distance that is similar in scale to that of recombination complexes.

Taken together, the above findings allow to specifically describe the interdigitated arrays of shorter and longer periodicity triad nucleation sites along pachytene chromosomes (Fig. 5E).

## Do longer periodicity triads arise at sites of minority crossover events?

Meiosis is known to give rise to two types of crossovers, the canonical type defined by Zip3 and Zip2 foci, plus a mysterious set of "minority" crossovers (Introduction). The results presented above reveal striking similarities between longer and shorter periodicity fluctuations in signal intensity. Since shorter periodicity triads arise at sites of canonical crossovers, a straightforward possibility would be that longer periodicity triads arise at sites of minority crossovers. Specific implications of this scenario would be that minority crossovers: (i) involve the same molecular triads, including ZMM components Zip2 and Zip3, as canonical crossovers; and (ii) are patterned as part of the same process that gives canonical crossover interference. This scenario is directly supported by two specific considerations.

(1)  Minority crossovers (total DNA-detected crossovers minus total Zip3 foci) and canonical crossovers (Zip3 foci) occur in a ratio of 0.46:1.0[18,31,32]. The ratio of longer periodicity triads to shorter periodicity triads is the same, 0.5:1, as given by the average ratio of longer to shorter periodicity iFFT peaks for Zip2 (0.5:1), Zip3 (0.6:1), Hop1 (0.5:1) and Zip1 (0.5:1) (Fig. 6A and Supplementary Fig. 11A).

(2)  Previous analysis of DNA-defined crossovers showed that total crossovers (canonical plus minority) exhibit reduced interference as compared to canonical crossovers[18]. To illustrate this point, we constructed CoC curves for total (DNA-defined) crossovers along budding yeast chromosomes IV and XV[18,31,32] (Fig. 6B). The revealed CoC relationships differ from those for canonical crossovers, in two related, diagnostic respects (compare Fig. 6B right panel versus Fig. 4B left panel. (i) Interference for total crossovers is much weaker than that for canonical crossovers alone ($L_{CoC}$ ~0.15 μm versus $L_{CoC}$ ~ 0.27 μm respectively. (ii) The CoC curve for canonical crossovers falls to zero at the smallest inter-interval distance, because interference ensures that crossovers rarely occur very near to one another along the same chromosome. In contrast, the CoC curve for total crossovers does not fall to a value of zero at the smallest inter-interval distances. Since interference is weaker, adjacent crossovers are now more often close together. These differences between the CoC curves for total crossovers and those for canonical crossovers reflect the effects of minority crossovers on the array of total events.

CoC relationships for total triads (shorter plus longer periodicity iFFT peaks) correspond very closely to those of total DNA crossovers (canonical plus minority) as seen for each of the three triad components and in the average for all components taken together (Fig. 6B–D and Supplementary Fig. 11B). Correspondingly, total triad CoC relationships differ from those for canonical crossovers (or shorter periodicity triads) in both diagnostic effects as described for total crossovers above. This outcome is a direct consequence of the patterns described above for the two types of triad patterns. Shorter and longer triad arrays are interdigitated. Interdigitation of two different periodic arrays will necessarily result in closer average spacing between adjacent events, and thus less interference, than seen for either array alone. More specifically, since longer periodicity peaks tend to occur immediately adjacent to shorter periodicity peaks, the total population of crossovers will exhibit a much higher frequency of chromosomes in which two adjacent events occur very close together (at shorter inter-interval distances) than for either type of triads or for canonical crossovers.

The findings presented above provide strong circumstantial evidence that longer periodicity triads arise at sites of minority crossovers. Put the other way around, we propose that minority crossover events nucleate longer periodicity triads. Based on this and other considerations, we propose that canonical and minority crossovers arise as two (likely sequential) effects of a single interference-governed ZMM-promoted recombination process (Discussion).

## Pch2/Trip13 modulates relative and absolute level of all triad components, specifically in longer periodicity triads

Previous cytological studies have shown that absence of Pch2 significantly alters the loading patterns of Hop1 and Zip1. In wild type,

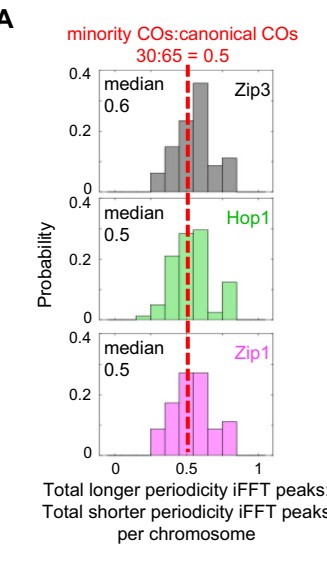
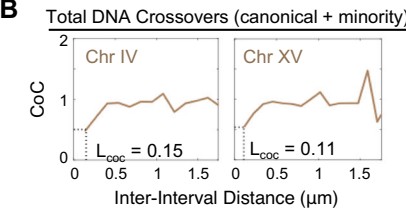
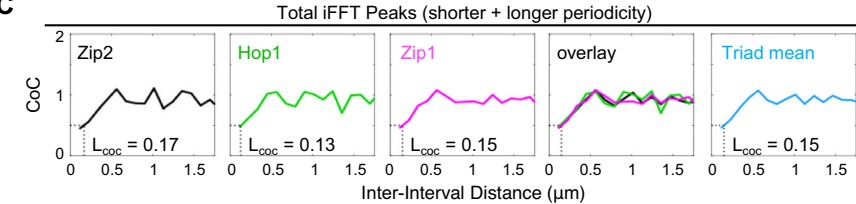
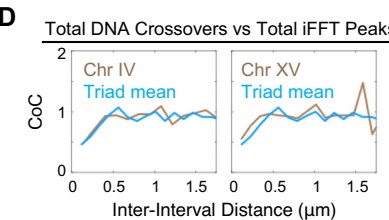

**Fig. 6 | Two lines of evidence that longer periodicity triads arise at the sites of "minority" crossovers. A** The ratio of total "minority" crossovers to total "canonical" crossovers is -0.46:1 (top, see text). The median ratios of total longer periodicity iFFT peaks to total shorter periodicity iFFT peaks is the same as for minority to canonical crossovers (average for all three components taken together = 0.53:1; red dashed line; *n* = 81 chromosomes). **B** CoC curves for total DNA-defined crossovers from published data[18,31,32] (*n* = 72 chromosomes for Chr IV and Chr XV). **C** CoC curves for total iFFT-defined peaks (shorter and longer periodicities considered together; *n* = 56 chromosomes) for each individual triad component and the mean CoC curve for all three components. **D** Comparison of CoC curves for total DNA-defined crossovers (from **B**) and mean for all three triad components (from **C**). Source data are provided as a Source Data file.

both molecules exhibit prominent long-range fluctuations in fluorescence signal intensity, with the levels of the two molecules often differing along the chromosomes (Introduction; Fig. 1E, F). In *pch2Δ*, signal intensity profiles seem to be smoother, with Hop1 and Zip1 present at much more similar relative abundances all along the chromosomes (e.g., Fig. 7A, B). Abundances of both chromosome-bound Hop1 and Zip1 are higher in a *pch2Δ* mutant than wild type *pch2Δ*[22]. We now see similar effects of *pch2Δ* on Zip3 fluorescent signal intensities (below).

To define the relative level of each different pair of molecules along each chromosome, we first defined their individual normalized signal intensities (as the number of standard deviations from the mean, e.g., Fig. 7C left) and then determined the (unsigned) difference in these normalized intensities at every pixel position (e.g., Fig. 7C right). The mean of these differences, which we refer to as the "dissimilarity index", summarizes the extent of differential relative abundance for that pair of molecules along that chromosome (e.g., Fig. 7C).

Along wild type chromosomes, dissimilarity is greatest for Hop1-Zip1, and smallest for Zip1-Zip3 signals (Fig. 7D black/orange, left). This is also true for the iFFT longer periodicity signal component (Fig. 7D black/orange, middle). In contrast, for the iFFT shorter periodicity component, all three pairs of molecules differ to a similar extent (Fig. 7D black/orange, right). In the absence of Pch2, dissimilarity is reduced for all pairs of molecules. Notably, this reduction is greatest for Hop1-Zip1, smallest for Zip1-Zip3 (Fig. 7D left, black/orange versus gray/blue), and is mirrored in the iFFT longer periodicity signal component (Fig. 7D middle, black/orange versus gray/blue), but not, or to a lesser extent, in the iFFT shorter periodicity signal component (Fig. 7D right, black/orange versus gray/blue),

iFFT outputs further reveals that longer periodicity triad peaks are substantially taller in a *pch2Δ* mutant as compared to wild type (e.g., Fig. 7E left versus right, blue). In contrast, shorter periodicity

iFFT triad peaks are very similar in the two cases (e.g., Fig. 7E left versus right, orange). Thus, in wild type meiosis, Pch2 reduces loading of triad components specifically, or primarily, in longer periodicity triads. We infer that this phenomenon underlies the previously documented total increases in abundance of chromosome-bound Hop1 and Zip1.

The above results show that, along wild type chromosomes, Pch2 operates specifically or primarily, upon longer periodicity signal components with two effects that together explain the visually apparent differences in fluorescence images of the three molecules along chromosomes isolated from wild type versus *pch2Δ* mutant cells. First, it actively increases the dissimilarity of the relative abundances of the three analyzed triad components. Second, it reduces the levels of all three components. Shorter periodicity signals, in contrast, are less or not affected by Pch2.

Since longer periodicity signals dominate total signal intensity patterns, these effects explain why *pch2Δ* mutant chromosomes exhibit brighter, smoother fluorescence intensity signals for Hop1, Zip1 and Zip3 (ref. 21; Fig. 7A versus Fig. 1D). It also implies that the prominence of shorter periodicity ("focal") signals, relative to longer periodicity ("domainal") signals will be reduced, as is the case of Zip3, whose focal signals are especially prominent in wild type (Fig. 7A versus 1D).

The original description of Hop1 and Zip1 loading patterns referred to "alternating domainal hyperabundance" in wild type (and its abrogation in *pch2Δ*)[21]. This description is sometimes taken to mean that Hop1 and Zip1 levels tend intrinsically to be anti-correlated. Quantitative analysis shows that this is not the case – the relative level of the two molecules in both shorter and longer periodicity triads varies over a wide but symmetrical range, with no indication of a bimodal distribution expected if loading levels tended to be anti-correlated (Supplementary Fig. 12A).

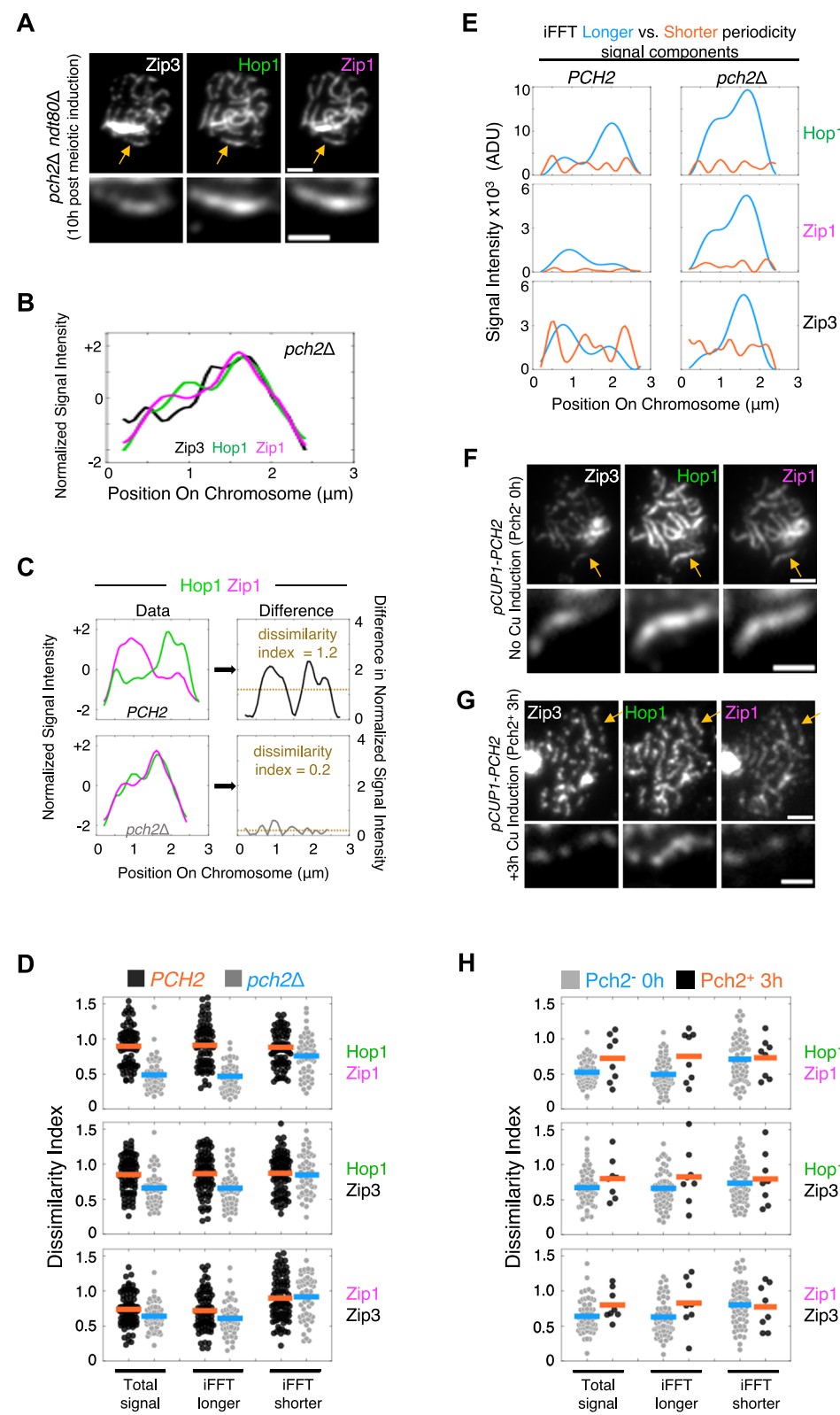

Additionally, previous work involving a bulk-population assay (chromatin immunoprecipitation) showed that Hop1 tends to be enriched at chromosome ends in a Pch2-dependent manner[33]. Our data show that Hop1 distributions vary considerably along both wild type and *pch2Δ* chromosomes without any reproducible pattern of terminal enrichment at the per-chromosome level (Supplementary Fig. 12B). However, when data for all chromosomes is averaged, a population-level tendency for Hop1 enrichment can be detected, with

the further suggestion that this effect is attributable primarily to effects of Pch2 on longer periodicity triads (Supplementary Fig. 12C).

### Induction of Pch2/Trip13 in a pachytene *pch2Δ* cell restores a wild type-like pattern of Hop1/Zip1/Zip3

In a *pch2Δ ndt80Δ* strain carrying a copper-inducible Pch2 expression construct (*pCUP1-PCH2*), arrested at pachytene in the absence of copper, chromosomes exhibit the typical cytological *pch2Δ* phenotype

**Fig. 7 | Pch2 modulates the relative and absolute abundances of different triad components, specifically in longer periodicity triads. A** Micrographs of immunostained spread chromosomes (*top*; scale bar = 2 μm) with example chromosome indicated by arrow (enlarged, *bottom*; scale bar 1 μm) from a pachytene arrested (*ndt80Δ*) *pch2Δ* cell. **B** Normalized signal intensity profiles of Zip3, Hop1 and Zip1 for the example *pch2Δ* chromosome in (**A**). Normalized signal intensity is defined as the number of standard deviations from the mean. **C** Differences in the relative levels of each pair of molecules along each chromosome was calculated as illustrated here for Hop1 and Zip1 along example *PCH2* and *pch2Δ* chromosomes (from Figs. 1F and 7B respectively). *Left*, normalized signal intensity profiles were first defined. *Right*, the (unsigned) difference in the two normalized signals was calculated at each assayed position. The mean of these values, referred to as the 'dissimilarity index', was then determined. **D** Distribution of dissimilarity indices, and their means (horizontal bars), for both the total signal, and iFFT longer and shorter

periodicity signal components, for all analyzed chromosomes in wild type (*PCH2*, *n* = 81) and *pch2Δ* (*n* = 52) samples. In *pch2Δ* versus *PCH2*, dissimilarity indices are significantly lower for all three pairs of molecules in both total signals, and longer periodicity signal components (*p* < 0.02, two-sided 2 sample t-test), but not significantly different for shorter periodicity signal components. **E** Comparison of signal intensity profiles for the three triad components reveals that the abundancies of all three molecules are increased in *pch2Δ* relative to *PCH2* in longer periodicity signals, but with no notable difference in shorter periodicity signals. ADU, analog-to-digital units. **F**, **G** As (**A**) except chromosomes are from inducible Pch2 cells grown in either the absence of Pch2 expression (**F**), or after subsequent expression of Pch2 for 3 h (**G**). **H** As (**D**) except analyzed chromosomes are from inducible Pch2 cells grown in either the absence of Pch2 expression (Pch2⁻ 0 h, *n* = 61), or after subsequent expression of Pch2 for 3 h (Pch2⁺ 3 h, *n* = 8). Source data are provided as a Source Data file.

(e.g., Fig. 7F as compared with Fig. 7A). If Pch2 expression is then induced and cells incubated for a further 3 h, chromosomes return to a more typical *PCH2* phenotype (e.g., Fig. 7G as compared with Fig. 1D). Correspondingly, the patterns of co-ordination in loading of different pairs of triad components before and after Pch2 induction match those of *pch2Δ* and *PCH2*, respectively (*n* = 61 and 8 chromosomes, before and after Pch2 induction, respectively; Fig. 7H as compared with Fig. 7D). Additionally, induction of Pch2 expression results in a significant reduction in the average signal intensities of all three molecules along pachytene chromosomes (Supplementary Fig. 12D). These are the outcomes expected if, upon induction, Pch2 carries out its above-defined roles in modulating the levels of components, specifically in longer periodicity triads. By extension, in wild type meiosis, Pch2 may have the potential to modulate the levels of triad components continuously throughout pachytene without affecting the existence of basic triad structure per se.

## Convergence of mathematical and biological findings

The results presented above illustrate the power of integrating mathematical analysis of periodicities with functional and biological considerations. FFT analysis defines two prominent fluctuations in signal intensity, of shorter and longer periodicity, which are the same for all three molecules. iFFT analysis reveals that the fluctuations of the three molecules are in phase, with resultant clustering of component peaks in well-separated ~200 nm triads. Biological data directly link shorter periodicity triads to sites of canonical crossovers. Biological data confirms the validity and functional significance of longer periodicity triads for recombination, by four types of observations. First, longer periodicity triads are affected by mutations known to affect interference for canonical crossovers. Second, longer periodicity triads are differentially affected by absence of Pch2. Third, the distance between a longer periodicity triad and the nearest shorter periodicity triad is the same as the known distance between adjacent recombination precursor sites. It seems unlikely that this is a coincidence; instead, it implies that longer periodicity triads arise at sites of recombination events, along the same set of precursors as canonical crossovers, in a particular (interdigitated) way. Fourth, the number of longer periodicity triads and their effects on crossover interference relationships correspond directly to those observed for minority crossovers, supporting a new explanation for these events.

## Discussion

The current study demonstrates that crossover interference sets up not one, but two, interdigitated, tiers of patterned events, of shorter and longer periodicity, both of which comprise closely clustered intensity peaks of crossover components (Zip2 and Zip3), axis component Hop1, and SC component Zip1. Shorter periodicity triads arise at sites of canonical crossovers. Put the other way around, canonical crossover interactions nucleate the formation of ensembles that contain not only crossover-specific ZMM components Zip2 and Zip3, as

defined previously, but also local accumulations of two structural components, Hop1 and Zip1. Diverse lines of evidence point strongly to the conclusion that longer periodicity triads arise at sites of minority crossover interactions, thus providing a cytological correlate for this type of event. Additional considerations suggest that both sets of events arise by a common ZMM-promoted pathway (further discussion below). We can propose two not-mutually-exclusive possibilities for the role(s) of triads.

- Model 1 (Fig. 8A): Meiotic recombination initiates by programmed DNA double-strand breaks (DSBs). The substrate for crossover designation is thought to comprise one (leading) DSB end in a nascent D-loop complex that is associated with the corresponding homolog partner axes while the other (lagging) DSB end remains associated with the axis of the originating homolog. At sites of canonical crossovers, crossover designation is likely imposed on the leading DSB end and, concomitantly, nucleates SC formation. The lagging end is brought into the developing complex only later, in "second end capture". We suggest that crossover designation triggers formation of a Zip2/3 assembly on the recombination complex per se; a Zip1 assembly on the leading end would stabilize its interaction with the SC; and a Hop1 assembly on the lagging end would allow eventual regulated release for second end capture, in accord with Hop1's intimate relationship to axis-associated cohesin complexes[34,35]. Similar roles could pertain at minority crossover sites.

- Model 2: Crossing-over requires not only reciprocal exchange between homolog chromatids at the DNA level but also creation of continuous chromosome structure at the corresponding positions ("axis exchange")[10], Fig. 8B. Crossover-specific recombination complexes and bits of SC remain colocalized at crossover sites during this process, with SC known to play a significant role[36]. Hop1 localization to these sites has not been examined, but might be suspected, where it could mediate separation of sister chromatid axes and/or promote development of new axis structure at these sites.

The presented findings suggest that the two types of crossovers, and their triads, arise at different positions along the same array of evenly spaced undifferentiated precursors sites, as governed by the same single basic interference process. An obvious way to ensure that the two sets of events occur at different positions along the same array is for them to arise sequentially, with canonical crossovers arising first and minority crossovers arising subsequently, at remaining unoccupied precursor sites. It is generally assumed that crossover interference is nucleated at the site of an event (of either type) and spreads out symmetrically in both directions, dissipating in strength with increasing distance. That spreading distance will be shorter for canonical crossovers and longer for minority crossovers, in accord with lesser and greater interference, respectively (Fig. 4C top versus Fig. 5A top).

Given these assumptions, a more specific scenario emerges (Fig. 8C). A chromosome would begin with a (near) uniform potential for crossover designation at all the precursor sites along its length. As

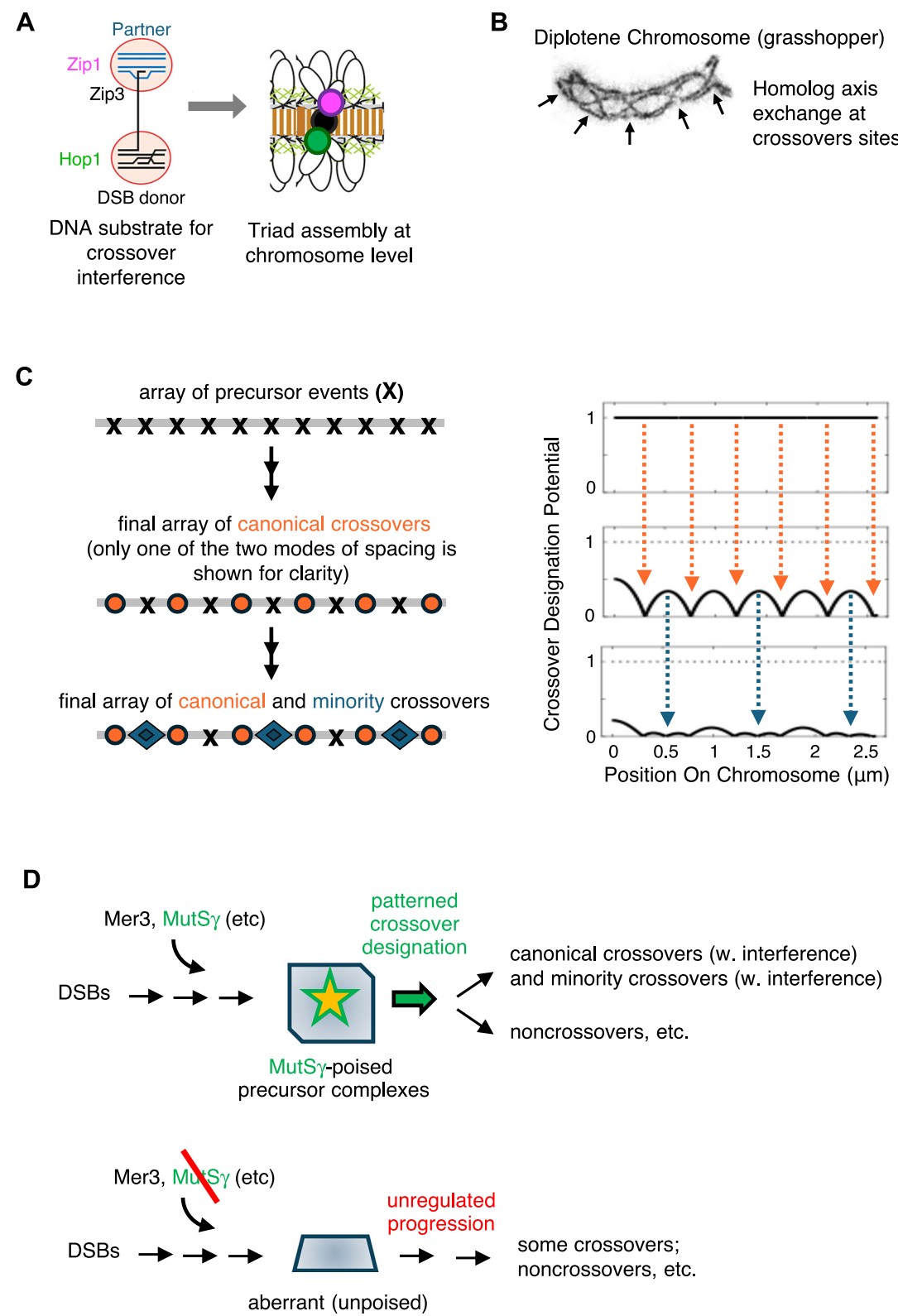

multiple canonical crossovers occur, they will create an uneven distribution of crossover designation potential which is lowest at the sites of the crossovers and increases with distance to either side. When minority crossovers then form, they will tend to occur at unreacted precursor sites with the highest residual designation potential, i.e., at precursors located between adjacent canonical crossovers. Since canonical crossovers are separated by either one or two intervening

precursors, minority crossover events will usually occur at a precursor site that is adjacent to either one or two canonical crossovers, in accord with observation (above). The second round of (minority) crossover designations would further reduce designation potential to levels too low to support further events.

Interestingly, the narrower and broader morphologies of canonical and minority crossover triads match the shorter and longer

**Fig. 8 | Two tiers of interference and roles of triads. A** Model for how triads might regulate DNA events involved with crossover recombination. DSB, DNA double-strand break (diagram adapted from ref. 10 under a CC BY 4.0 license). (**B**) Electron micrograph of grasshopper late meiotic prophase (diplotene) chromosome with interhomolog axis exchanges indicated (adapted from ref. 58 with permission from Cambridge University Press through PLSclear). **C** Two tiers of triads, of shorter and longer periodicity, can be explained by two sequential rounds of crossover interference. The two rounds are implemented on the same array of (evenly spaced) undifferentiated precursor interactions with full potential to undergo crossover designation (top). (For simplicity, all interactions are shown as having the same potential, although this is likely an oversimplification[18,28]). A first round of crossover designation (orange arrows) gives shorter periodicity (canonical crossover) events

separated by two or three inter-precursor distances, with concomitant reductions in crossover designation potential in the region surrounding each event (middle). A second round of crossover designations (blue arrows) occurs at unreacted precursor positions with the highest remaining potential, which are necessarily located between adjacent first round events, resulting in further reduction in designation potential (bottom). Crossover designation potential could correspond to the level of mechanical stress along axes as in the proposed "stress hypothesis"[10,37]. **D** *Top*, the same array of precursor complexes, poised for crossover designation in a manner dependent upon MutSγ, could give rise to both canonical and minority crossovers via two rounds of interference as in (**C**). *Bottom*, in the absence of MutSγ, precursor complexes develop but are not arrested in a poised state and thus progress without regulation to give residual crossovers that lack interference.

distances over which the effects of interference spreads. Perhaps spreading of triad components outward from a nucleation site is limited by the level of surrounding "designation potential". In the first round of events, designation potential rises steeply to either side of the crossover site, limiting spreading; in the second round, designation potential is reduced all along the chromosome, permitting broader spreading.

The features described in this two-stage scenario can easily be explained if crossover designation is the result of mechanical stress, with resultant reduction and redistribution of stress accounting for interference[18,37,38]. It could equally well be explained by any model that follows similar logic. However, it is not explained by current versions of the recently proposed coarsening model[39]. In that model, relevant components move from smaller assemblies on a larger number of precursor sites into larger assemblies on a smaller number of canonical crossover sites, in a single continuous process. Without additional assumptions, it does not allow for two interdigitated evenly spaced arrays governed by different "interference distances".

As defined in budding yeast, "minority" crossovers are events that occur in wild type cells and are detected as an excess of DNA-detected events over cytologically defined "canonical crossovers" (Zip2/Zip3 foci). We present evidence above that longer periodicity triads correspond to these minority events. Notably, the properties of longer periodicity triads and their relationship to canonical crossovers are at odds with current dogma regarding the nature of minority crossovers. Longer periodicity triads: (i) exhibit interference, which is even stronger than that of canonical crossovers; and (ii) are marked by the same molecules that mark triads at canonical crossover sites, including ZMM proteins Zip2 and Zip3, raising the possibility that they might also be promoted by ZMM components. In contrast, current dogma, across diverse organisms, holds that minority crossovers in wild type meiosis: (i) do not exhibit interference; and (ii) are ZMM-independent, because they are same as the residual crossovers that occur in the absence of ZMM components MutSγ or Mer3 (and which do not exhibit interference), e.g., refs. 40–43. However, reconsideration of published data suggests that all prior observations, across different organisms, can be understood in the context of the conclusions presented here.

- In yeast, modeling studies showed that the distribution of total crossovers (canonical plus minority) could be explained if non-interfering minority crossovers were "sprinkled randomly" among canonical crossovers[44]. In maize, the existence of two different types of pachytene nodules was explained analogously[45]. However, the current study presents experimental data for yeast (and thus equivalently for maize), that total crossovers distributions are explained quantitatively by two tiers of interfering crossovers. The modeling studies do not directly detect the second tier of interference per se. They give acceptable results essentially because the minority of second tier crossovers are interdigitated among canonical (first tier) crossovers, with interference acting over a very long distance.

- A unique study in tomato identifies two types of recombination nodules in electron microscopy images[46]. 82% exhibit MHL1

fluorescence staining, exhibit classical interference, and thus correspond to "canonical" crossovers. 18% of nodules lack MLH1 staining and are inferred to correspond to "minority" crossovers. Interference among these MLH1-negative nodules could not be directly assessed because they are too few in number. However, several features are described, all of which are predicted by occurrence of two successive tiers of crossover interference as outlined in Fig. 8C. (i) In tomato, the two types of nodules do not occur independently but instead "interfere" with one another. This interplay occurs in the two tiers model because minority crossovers tend to occur roughly midway between canonical crossovers, where remaining crossover designation potential is highest. This is seen in tomato as a lower frequency of MLH1-negative nodules close to MLH1-positive nodules than expected if the two types occurred independently, with interference extending farther between pairs of MLH1-positive nodules than between mixed pairs of positive and negative nodules. (ii) The numbers of the MLH1-positive and MLH1-negative nodules are anti-correlated on a per-chromosome basis. This is directly predicted by the two tiers model: regions with more canonical crossovers will have less residual crossover designation potential, and thus fewer second tier/minority crossovers, than regions with fewer canonical crossovers (and vice versa). (iii) MLH1-positive nodules arise earlier in meiosis than the MLH1-negative nodules, in accord with sequential occurrence of first and second tier triads (i.e., canonical and minority crossovers). (iv) Finally, it was inferred that the MLH1-negative nodules do not exhibit interference based on the fact that there are many chromosomes without even one such nodule, thus giving a Poisson-like distribution of the number of nodules per chromosome. The two tiers model can explain this finding. For canonical crossovers, the "obligatory crossover" rule ensures occurrence of at least one canonical crossover on each bivalent[8,10]. However, during second tier events, chromosomes with many/more first tier crossovers may not have sufficient residual crossover designation potential to give rise to a second-tier event, with a resulting high level of zero-minority crossover bivalents.

- We further note that, in the tomato study above, MLH1-positive and MLH1-negative nodules are very similar, discrete structures as visualized by EM. This correspondence suggests that the earlier events that give rise to the two types of complexes are quite similar in the two cases. This similarity matches the current observation that longer and shorter periodicity signals exhibit similarly sized local (nucleating) triads that contain the same molecules (including ZMM proteins).

- Conflation of residual *mutSγ* and *mer3* crossovers with minority crossovers in wild type rests on the assumption that these mutations simply subtract the canonical "ZMM-mediated" pathway, leaving behind some other process that also occurs in wild type cells and does not exhibit interference. However, there is no direct evidence for this interpretation. Moreover, in mouse, mutations of MutSγ that affects the ATP binding domain confers a complete absence of all crossovers, implying that both minority and canonical crossovers are under the influence of this ZMM protein[43,47]. Thus, we propose that MutSγ and Mer3 are required for development of undifferentiated precursor complexes that are poised in a state where they are susceptible to

interference-mediated crossover designation (Fig. 8D top). In this scenario, in the absence of MutSγ or Mer3, recombination will proceed but without such regulation, with resulting crossovers lacking interference (Fig. 8D bottom). And in mouse, in the absence of ATP binding or hydrolysis, assembly of the complex can occur but failure of turnover results in failure of progression of (crossover) recombination complexes at this and/or later stages.

- Finally, we note that canonical crossovers are thought to be resolved by Mlh1/3 while minority crossovers in wild type are thought to be resolved by Mus81/Mms4. Evidence in budding yeast is not consistent with differential activities of the two types of resolvases. Evidence consistent with such a possibility has been presented for Arabidopsis and tomato, but with caveats (Supplementary Note).

In current dogma, canonical crossovers are referred to as Type I, while minority crossovers in wild type and residual crossovers in *msh4/5* mutants are both referred to as "Type II". Since the latter two types of crossovers seem not to be the same, the terms "canonical", "minority" and "residual" crossovers may be more appropriate.

We further note that the AAA+ -ATPase Pch2 is not involved in setting up triad structure. However, it does affect relative levels of the analyzed components. The current data show that absence of Pch2 results in a reduction in the differences in levels of different components, and that this effect occurs specifically in longer periodicity triads. Thus, Pch2 acts specifically in longer periodicity triad signals to uncouple the levels of different components. In shorter periodicity triads, different components occur at different levels irrespective of Pch2. Absence of Pch2 is also known to increase the abundance of chromosome-bound Hop1 and Zip1[22]. We find that Zip3 is affected in the same manner (Supplementary Fig. 12D). Together these effects can account for previously described differences between wild type and *pch2Δ* chromosomes. The observed effects likely involve multiple, complex interactions among the involved components: Pch2 can act directly on Hop1 to promote both its loading onto and removal from chromosomes[34,48]; Pch2 is only seen loaded onto chromosomes in the presence of Zip1[49,50]; and Zip1 is known to promote re-localization of Zip3[26].

An interesting finding from the current work is that Pch2 can act dynamically to restore differential signal intensities even after the Pch2-independent condition has been established. This finding raises the possibility that analogous dynamic effects can occur during, and/or prior to, the pachytene stage in wild type meiosis.

## Methods

### S. cerevisiae strains
All strains used in this study are *MATa/MATα* derivatives of SK1 and are listed in Table 1.

### Meiotic induction
Mitotically growing cells were induced to undergo synchronous meiosis at 30 °C using the previously described SPS method[51]. Time 0 was defined as the transfer of cells to sporulation medium. For *NDT80* strain LZY819, cells were harvested for chromosome spreading 6 h later, when pachytene cells were most abundant. For *ndt80Δ* strains BWY588, LZY2431, and BWY595; cells were harvested 10 h post meiotic induction for maximal accumulation of pachytene cells. For *pCUP-PCH2 ndt80Δ* strain BWY579, a sample of cells ('No Copper induction') was first harvested 10 h post meiotic induction. Copper was then added to the growth medium to induce *PCH2* expression and a second sample of cells 3 h later (13 h post meiotic induction).

### Preparation of chromosome spreads from harvested pachytene cells
Chromosome spreads were prepared essentially as described in ref. 52. Harvested cells were spheroplasted to remove their cell wall before resuspending in MES wash (1 M sorbitol, 0.1 M MES, 1 mM EDTA, 0.5 mM MgCl₂ pH 6.5). Cells were then lysed with 1% Lipsol detergent, before spreading their contents onto a glass microscope slide and adding a fixative (3% w/v paraformaldehyde, 3.4% w/v sucrose).

### Immunostaining spread pachytene chromosomes
Immunofluorescent labeling was performed as previously described[53]. Slides containing spread, fixed pachytene chromosomes, were incubated at room temperature in TBS buffer (25 mM Tris-Cl, pH 8, 136 mM NaCl, 3 mM KCl) then blocked with TBS buffer-1% w/v Bovine serum albumin (BSA). Primary antibodies (mouse monoclonal anti-Myc for Zip3 (Santa Cruz Biotechnology; catalog# sc-40; lot# H1721), goat polyclonal anti-Zip1 (Santa Cruz Biotechnology; catalog# sc-15632) rabbit polyclonal anti-Hop1 (generous gift from Franz Klein, Max Perutz Labs, Vienna), or rat polyclonal anti-Zip2 (custom antibody, Abclonal; custom antibody, project# AP20721) were diluted 1/1000 in TBS- 1% BSA.

For *ndt80Δ* experiments involving labeling of Zip3, Hop1 and Zip1, secondary antibodies were donkey anti-mouse, anti-rabbit, and anti-goat IgG labeled with Alexa555 (for Zip3; Invitrogen; catalog# A-31570; lot# 1117032), Alexa488 (for Hop1; Invitrogen; catalog# A-21206; lot# 2289872), and Alexa647 (for Zip1; Invitrogen; catalog# A-21447; lot# 1739289) and diluted 1/1000 in TBS- 1% BSA. For *NDT80* experiments involving labeling of Zip2, Hop1 and Zip1, a distinct set of secondary antibodies were used. Secondary antibodies were donkey anti-rat, anti-rabbit, anti-goat labeled with Alexa488 (for Zip2; Invitrogen; catalog# A-21208; lot# 2310102), Alexa555 (for Hop1; Invitrogen; catalog# A-31572), and Alexa647 (for Zip1; Invitrogen; catalog# A-21447; lot# 1739289). These secondary antibodies were also used at a dilution of 1/1000 in TBS- 1% BSA. Chromosomal DNA was stained with DAPI (1 µg/ml) to permit detection/localization of pachytene chromosomes without the risk of bleaching the immunofluorescence signals of interest. Finally, the labeled chromosomes were mounted in Prolong Gold (Invitrogen Molecular Probes).

### Materials availability
Yeast strains and antibodies can be obtained from the corresponding author upon request.

### Image acquisition
Micrographs were obtained by widefield fluorescence microscopy using a Zeiss Axioplan 2ie MOT microscope, with an attached plan-apochromat 100x magnification, 1.4 numerical aperture, oil immersion objective (Zeiss), and a Hamamatsu ImageEM EM-CCD camera (model C9100-13; effective pixel size, 67 nm). The microscope was controlled by Metamorph (Molecular Devices) software.

### Obtaining signal intensity profiles of spread pachytene chromosomes
Signal intensity profiles of pachytene chromosomes were extracted from acquired images using open-source software FIJI[54] (ImageJ v1.54 f). Isolated chromosomes were traced along their mid-line of the Zip1 image using the segmented line selection tool with a 3-pixel width (Fig. 1D bottom). This region of interest was then copied onto the corresponding Hop1 and Zip3 images. The signal intensity values of each all three regions of interest were extracted using the Plot Profile function of FIJI and saved as.csv files for downstream analysis. Images were not pre-processed prior to measurement. A simulated chromosome-like object (curvilinear line whose length and width matched measured chromosomes) with diffraction-limited spots of known pixel intensity values was used to validate this approach (Supplementary Fig. 2A). A 3-pixel width line was selected to capture the majority of the chromosome signal while minimizing contribution of extra-chromosomal pixels. However, control analysis showed that results were robust to the line thickness used (Supplementary Fig. 2B).

**Table 1 | *S. cerevisiae* strain list**

| Strain Name | Genotype | Corresponding Figure(s) | Source |
|---|---|---|---|
| NKY4160 (BWY588) | *ho::hisG/", ura3/", ndt80Δ::LEU2/", ZIP3-13myc::HphB/ZIP3, leu2::LEU2:tetR-GFP/", lys2::TetOx240:URA3/LYS2* | Figs. 1–7 and Supplementary Fig. 1 – 4, 6, 7, 11, 12 ('*ndt80Δ*', '*PCH2*', 'Zip3') | This work |
| NKY4147 (LZY819) | *ho::hisG/", leu2::hisG/", ura3(SmaI-PstI)/", ZIP3-13myc::HphB /", URA3::CYC1p-LacI-GFP/", scp1::LacO-LEU2/", V5-SMT3::Blasticidin/?* | Fig. 6 and Supplementary Fig. 5, 7, 9, 10, 11, ('*NDT80*', '*Zip2*') | Gift from Lian-gran Zhang |
| LZY2431 | *ho::hisG/", leu2::hisG/", ura3(SmaI-PstI)/", pCLB2-TOP2::kanMX/" ndt80Δ::kanMX4/", ZIP3-13myc::HphB /", URA3::CYC1p-LacI-GFP/", scp1::LacO-LEU2/",* | Figs. 4, 5 and Supplementary Figs. 7, 8 ('*top2*') | Gift from Lian-gran Zhang |
| NKY4161 (BWY595) | *ho::hisG/", ura3/", pch2Δ::kanMX/", ndt80Δ::LEU2/", ZIP3-13myc::HphB/ZIP3, leu2::LEU2:tetR-GFP/", lys2::TetOx240:URA3/LYS2* | Figs. 4, 5, 7 and Supplementary Figs. 8, 12 ('*pch2Δ*') | This work |
| NKY4162 (BWY579) | *ho::hisG/", ura3/", pCup1-3HA-PCH2::kanMX/", ndt80Δ::LEU2/", ZIP3-13myc::HphB/ZIP3, leu2::LEU2:tetR-GFP/", lys2::TetOx240:URA3/LYS2* | Fig. 7 and Supplementary Figs. 12 ('*pCUP1-PCH2*', 'Pch2-', 'Pch2+') | This work |

Footnote. Italics indicate genotypes.

**Determination of the two most prominent periodicities (Fig. 2A–C, and Supplementary Fig. 5A).** The Fourier transforms of all signal intensity profiles (e.g., Fig. 2A) were calculated using MATLAB's fft function to produce one-sided amplitude spectra (e.g., Fig. 2B). For each amplitude spectrum, the two peaks (local maxima) with the highest amplitude were determined using MATLAB's findpeaks function (default parameters; e.g., Fig. 2B asterisks). Distributions of these values, for all signal profiles, of all measured chromosomes were fit to two component Gaussian mixture models using MATLAB's fitgmdist function (e.g., Fig. 2C).

**Inverse Fast Fourier Transform (FFT) analysis**
The Fourier transforms of all experimental signal intensity profiles (e.g., Fig. 2A) were first calculated using MATLAB's fft function (as above). Next, we specifically isolated the two groups of frequencies above and below the desired threshold (875 nm). For the shorter periodicity group, amplitudes of frequencies above the threshold were set to zero; for the longer periodicity group, amplitudes of frequencies below the threshold were set to zero. For each of the two resulting groups of amplitudes, MATLAB's inverse iFFT function was then used to convert the Fourier transforms (in k-space) back to position space (i.e., into signal intensity profiles in units of micrometers along chromosomes), e.g., Fig.3A.

**Summing all Fourier Transforms (Fig. 2D, E and Supplementary Fig. 6).** For each analyzed molecule: (i) The primary experimental signal intensity profiles for all chromosomes in the data set were normalized by setting the largest intensity in each profile to a value of 1, thus ensuring that each chromosome would contribute equally to an eventual sum. (ii) We then calculated the discrete Fourier transform of each experimental normalized signal intensity profile using the standard Mathematica software command Fourier. The absolute values of those Fourier transforms represent the amplitudes of the different frequencies as a function of the Fourier wave vector "k". (iii) To accommodate the fact that different chromosomes have different lengths, we interpolated these k-space Fourier transforms so that they all had the same k-space range. (iv) We then summed all of the individual interpolated k-space Fourier transforms. (v) Finally, we converted this sum from k-space to position space, using the relationship that each k vector periodicity corresponds to a characteristic wavelength $l = 2Pi/k$ and including transformation of the bin widths from k-space to position space. The resulting graphs thus show the

distribution of probabilities that the data set includes periodicities of different lengths (l) in μm, with the highest probabilities representing the most prominent periodicities. The final distributions were then normalized by setting the height of the largest value to unity (1) for experimental data (Fig. 2D) and to 0.95 for simulated data (Fig. 2E; Supplementary Fig. 6).

**Detection and downstream analysis of shorter and longer periodicity iFFT-generated peaks**
Shorter and longer periodicity peaks were detected in iFFT outputs using custom MATLAB function getSignalPeakAndHumpPositions. This function uses the findpeaks function of MATLAB (default settings) at its core to detect local maxima. Occasionally, peaks are missed by this approach as they appear as 'humps' rather than local maxima. Therefore, to detect these 'peaks', this function scans for any humps that were not detected by searching for local maxima (using the findpeaks function of MATLAB) on the first derivative of the signal. Prior to application of peak finding, to minimize noise detection, the smooth function of MATLAB was utilized to smooth shorter periodicity signals with a moving average filter of 3 pixels. Distances between adjacent shorter/longer periodicity peaks were calculated using custom MATLAB function getCOSpacing. Distributions were fit to either single component gamma distributions using the gampdf function of MATLAB, or two component Gaussian distributions using the fitgmdist function of MATLAB. Custom MATLAB function getDomainalToFocalPeakDistances was used to calculated distances between each longer periodicity iFFT peak and its nearest adjacent shorter periodicity iFFT peak. Custom MATLAB function getDistanceToClosestPeak was used to calculate (closest) distances between triad components. Coefficient of coincidence values were calculated using custom MATLAB function getCoC as previously described[28]. 30 intervals were used for analysis of both shorter periodicity peaks and total peaks (shorter periodicity plus longer periodicity peaks), whereas 17 intervals were used for analysis of longer periodicity peaks.

**Analysis of genetic crossovers**
Published genetic crossover positions for yeast chromosomes IV and XV were converted from base pairs DNA into μm chromosome length assuming 324 bp/nm. This conversion was based on the known length of these chromosomes (1.53 Mb and 1.05 Mb respectively) and measured average pachytene lengths (4.8 μm and 3.2 μm[18]), which give

conversion factors of 319 nm/bp and 328 nm/bp respectively. 324 bp/nm is the average of these two values.

## Dissimilarity Index (Fig. 7C, D and H)

The signal intensity profiles of each molecule were normalized using custom MATLAB function normalizeSignalsBySTD which first subtracts the mean signal intensity of the profile and then divides by the standard deviation of the profile. Then, for each chromosome, the (unsigned) difference in normalized intensity was calculated for each pairwise combination of triad components at each position along the chromosome using custom MATLAB function getTriadSignalDifferences. We refer to the mean of these per-chromosome values as the 'dissimilarity index'.

## Population average signal intensity profiles (Supplementary Fig. 12C)

Population average signal intensity profiles of each molecule were obtained using custom MATLAB function getPopnAverageSignal. Briefly, each measured chromosome was first normalized to unit length by dividing each pixel position (in micrometers) by the chromosome length (in micrometers). To permit averaging, chromosomes must have a signal intensity measurement at the same normalized chromosome position as one another. This was achieved by spline fitting followed by interpolation.

## Reporting summary

Further information on research design is available in the Nature Portfolio Reporting Summary linked to this article.

## Data availability

All primary data have been deposited as dataset "Crossover Interference Mediates Multiscale Patterning Along Meiotic Chromosomes to the Harvard Dataverse database (https://dataverse.harvard.edu) at https://doi.org/10.7910/DVN/5LEWYF[55]. The underlying numeric data for Figs. 1F, 2A–E, 3A–F, 4B–E, 5A–C, 6A–C and 7B–D, E, H is in the Source data file. Source data are provided with this paper.

## Code availability

The analysis code that was generated in this study is publicly available via the Github repository (https://github.com/mwhite4/multiscaleCrossoverPatterning)[56].

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

## Acknowledgements

We thank Kathleen Fleming and Fengfeng Zhuang for preliminary experiments; John Hutchinson for advice on stress-promoted patterning; Maria Mukhina for help with Fourier analysis; Liangran Zhang for yeast strains, and Franz Klein and Nancy Hollingsworth for anti-Hop1 antibodies. This work was funded by National Institute of Health grant R35GM136322 (N.K.), Human Frontiers Science Program long-term fellowship LT000927/2013 (M.A.W.), and the Chu Family Foundation (M.P.).

## Author contributions

Conceptualization, M.A.W., B.W., and N.K.; Methodology, M.A.W., B.W., M.P., and N.K.; Software, M.A.W. and M.P.; Formal Analysis, M.A.W. and M.P.; Investigation, B.W., L.C., and G.L.; Resources, B.W. and N.K.; Writing–Original Draft M.A.W. and N.K.; Writing–Review and Editing, M.A.W., B.W., G.L., M.P., and N.K.; Visualization, M.A.W.; Supervision, N.K.; Funding Acquisition, N.K., M.A.W., and M.P.

## Competing interests

The authors declare no competing interests.
