## [Transparent Peer Review file · Nature Communications]

Crossover Interference Mediates Multiscale Patterning Along Meiotic Chromosomes

Corresponding Author: Professor Nancy Kleckner

Version 0:

Reviewer comments:

Reviewer #1

(Remarks to the Author)

White et al. describe an analysis of immuno-labeled chromosome spreads of yeast nuclei, primarily, but not exclusively, in the *ndt80Δ* arrest. They trace the label along chromosome midlines and find that the observed patterns contain 2-3 periodic components. By separating short from long wavelengths through Fourier analysis, they find that the short wavelengths largely fit what is in cytology referred to as 'foci', while the longer wavelengths correspond to what was described as 'domains' of the corresponding proteins. Strikingly, both foci and domain patterns co-localize among three key components of meiotic chromosomes: Hop1, Zip1, and Zip3. Foci patterns arguably correspond to majority COs, since they align with Zip3, while the domain patterns fit those of 'minority' COs.

In general the paper is well written and performed with great insight.

The approach to trace chromosome signal and employ a periodicity detector such as Fourier-analysis is interesting.

It is challenging to critically understand the work. I think that the analysis shows important points.

My comments below aim to clarify issues and to help to engage the readership in critically retracing reasoning and results.

i) Tracing chromosomes - pixelation.

Yeast chromosomes are small, even in pachytene.

When analyzing or measuring linear features in images, especially when using a limited number of 3 pixels for the width of the tracing, the shape and orientation of the object can significantly affect the measurement. The authors are very likely aware of the problems. It would nevertheless be helpful to address these problems briefly, so as to keep the audience aware.

I have illustrated some obvious pixelation issues in a sketch below.

a) Changing the relative orientation of the same object relative to the pixel grid will affect its representation

The object is perfectly aligned with the pixel grid in A - but rotated by 45° in B. Below the object, I plotted how the object is likely represented after pixelation - intensities are now distributed over a larger set of pixels with reduced intensities.

Thus: Straight objects can be represented differently depending on their alignment with the pixel grid.

b) In addition the 'midline' (14 pixels in A) is represented by only 10 pixels in B, due to their diagonal arrangement.

The question arises, how a "width of 3 pixels" is realized under the circumstance of a 45° rotation.

It's difficult because, due to the reduced number of pixels in the midline, the number of adjacent pixels may also be reduced, when at the same time the label is smeared over more than the original 28 pixels.

c) kinks and bends

Objects, which are bent or have kinks of various degrees, may compress or decompress label accordingly, possibly causing fluctuations in what might otherwise be a flat intensity in the straightened state (C).

c) The problems may be aggravated when chromosomes with kinks or bends (C) are arranged arbitrarily relative to the grid.

Suggestion: It would be great to comment in the text, why these issues do not affect the Fourier outcomes. In the supplement or computational methods section, one might explain how the algorithm deals with pixelation. If one had a graphical representation, that would, of course, be of the highest didactic value.

ii) Tracing chromosomes- selection

Based on the strain table, it is possible that specific chromosomes (II and XV) were selected for tracing. If there was any selection, it would be beneficial to add this description to Figure 1. Given that three channels were used for the three proteins, it is unclear whether a channel was available to recognize a chromosome. If chromosomes of different lengths were traced, were always all chromosomes of a spread traced, or only those that appeared continuous? Or were chromosomes with a minimal length selected?

If chromosome lengths varied, a comment on the effect of the length distribution on the resulting Fourier coefficients would be highly appreciated. In cases where specific chromosomes were analyzed, it would be beneficial to announce this prominently (e.g., Fig. 1).

ii) Fourier analysis - PSD

For readers who are not very familiar with Fourier analysis, the magnitude plot may be more accessible than the power spectrum density (PSD). PSD is essentially the square of the magnitude and is often used when the result is interpreted as the “energy” of the underlying frequencies. The coefficients of magnitude could be interpreted as relative contributions of the respective frequencies to the signal, which is fairly intuitive. Of course, the authors should use what they think is most appropriate.

ii) Fourier analysis - arguments/binning

The PSD plots are presented as semi-log plots, where the arguments are plotted linearly and the PSD coefficients in log scale. However, it appears that for the arguments bins of exponentially decreasing size were used, resulting in a very crowded lower left part. It would be beneficial if the complete list of arguments could be presented (such as the law that generates them) together with the rationale for choosing this particular set.

If a magnitude plot were presented in a log-log format, this would decluster the left corner and the reader might be able to interpret the coefficients as relative contributions of frequencies to the signal. In its current form, it is very difficult to interpret the values due to the squaring and the clustering of values in the left part.

ii) Fourier analysis - two ‘most prominent’ peaks

It might be helpful for the reader if the criterion for selecting the two ‘most prominent’ peaks in the PSD were stated in the figure legend or text close to its first occurrence. For instance, stating “the two local maxima with the longest periodicities were chosen” would be consistent with the six peaks shown in Fig. 2A. In the absence of thresholding, this step produces unstable values, which the authors compensate for with statistical analysis over the complete dataset.

ii) Fourier analysis - statistics of power density peaks

The $n \times 2$ peak periodicities (per protein and chromosome) are approximated by a combination of three Gaussian curves. My feeling is that the densities are noisy, especially for the longer frequencies, potentially signaling that the collected data are on the lower limit. For example, for the important Zip3 density, it is not obvious that it is best approximated by ‘three Gaussians’.

While the analysis of the data, as it currently stands, yields reasonably robust results after fitting, it would be fantastic to actually see the three Gaussian peaks emerge also for Zip3, which is currently too noisy to show three peaks. To make sure the “three Gaussian” model is really justified, Fig. 2 might benefit from a 2-3 fold increase in the data set.

If the three peaks show clearly for each protein in one data set (i.e., Fig. 2), the other data sets could be approximated in the same way— even with more sparse data.

ii) Fourier analysis - inverse FFT

In this analysis, frequencies $< 600\text{nm}$ and frequencies $> 900\text{nm}$ were used to define high and low frequency components of the signal. Strikingly, for this example, the high frequency patterns of all three proteins show convincing colocalization. On this occasion, the authors may comment that they checked for ‘bleed through’ in their three channels, which could also lead to striking colocalizations. For orientation, it would be beneficial to plot the sum of the two inverse FFTs ($<600, >900\text{ nm}$) as a dashed line in the top panel of Fig. 2C, to demonstrate how well the two frequency ranges capture the original profile and confirm that nothing important was discarded.

ii) Colocalization of high frequency peaks ('foci')

These data are quite clear. Please indicate somewhere prominent, e.g. in the legend, that this statistics is for the complete data set and not only for the example chromosome.

iii) Coc

Coc: It may be helpful to repeat the Coc formula (as in Zhang, 2104, Fig. 1 c) in the figure, mentioning in the legend that chromosomes were split into 30 (foci) or 15(domain) equal intervals.

The other results and the discussion follow smoothly from the discussed parts.

(Remarks on code availability)

Reviewer #2

(Remarks to the Author)

The manuscript by White, Kleckner and colleagues carefully examines the distribution of three key factors along meiotic chromosomes in order to understand the mechanism underlying the distribution of crossovers. This is an important and timely question. The patterning of crossovers has implications for a wide array of fields, from basic chromosome biology to infertility to evolution. Crossovers are uniquely positioned in every meiocyte; however, they exhibit intriguing patterns of co-occurrence, both at the population level and relative to one another in every single meiosis. This work is timely since recent work on condensates led to the reexamination of several of our assumptions about the mechanisms that pattern crossovers. In addition, advances in microscopy techniques and genetic analysis in multiple organisms allowed us to gain unprecedented knowledge of meiotic chromosomes.

The authors revisit a cytological approach to localize meiotic structure on budding yeast meiotic chromosomes. This is an appropriate method to study a phenomenon that is 'single-molecule' in nature. They develop an automated analysis pipeline using minimally processed images to reveal difficult-to-discern patterns of distribution. They confirm this approach by recapitulating previous observations of the regularity of crossover positioning but also reveal previously undocumented 'domainal' periodicities. This analysis leads to the conclusion that a large-scale periodicity reflects 'minority' crossovers (also called non-interfering or class II crossovers). These events, which represent a minority of exchanges, are poorly understood, so defining a cytological correlate for them would be of high significance. They then integrate their findings to model a two-step patterning process that can recreate empirical distribution of crossovers.

The conclusions of this work, if substantiated, would be of high interest to the meiosis and chromosome biology communities. Given budding yeast's centrality as a model organism for meiosis, this work would be of wide interest to researchers working in all model organisms. However, multiple potential issues related to image acquisition, mathematical analysis, statistical testing and lack of empirical testing of the proposed hypotheses cast doubt on the conclusions. Below are some of the most crucial issues:

1. As the authors themselves claim, the 'fundamental findings of this study' rely on fluorescent micrographs. The authors test two potential artifacts in these images - two different proteins that localize to future exchanges (Zip2 and Zip3) and both physiological and arrested (*ndt80Δ*) meiocytes. However, a few other controls are essential. First, the observed effects are on a similar size scale as both the pixel size (67nm; 3-pixel averaging brings it to close to half of the observed focal periodicity, and even more than half of the focal periodicity in some mutant scenarios; it also happens to be the hypothesized spacing between precursors) and the diffraction limit of fluorescence microscopy (~150-200nm). It is essential to confirm that these technical aspects do not influence the analysis. Some of the possible solutions are to acquire and analyze similar images using a super-resolution microscopy system with different resolution and pixel size, and to change pixel size in this system (e.g., by adding an optovar). Second, the minor amplitude of some of the effects (e.g., the 'focal' peaks in the Hop1 & Zip1 data), raises worries about bleed-through between the fluorescent channels. This should be tested by swapping secondary antibodies and repeating the analysis. A final suggestion is not a requirement but would be of interest. Advances in microscopy should enable analysis of chromosomes in structurally preserved samples, rather than the chromosome spreads used here. Given the large body of work from the Kleckner lab and others on this system, it would be beneficial to confirm the existence of these patterns in multiple sample preparation protocols, ideally including minimally perturbed ones.

2. The conclusions relating to the correspondence of domainal triads to class II crossovers are premature. They are based on mere correlations, and are not tested. Without a way to substantiate these correlations, this conclusion should be moved to the Discussion and clearly be marked as speculative. (See point 3, however.)

3. The modeling of crossovers in the beam-film framework as two waves with different properties is intriguing. By itself, its 'economic' nature does not significantly add to, or validate, any of the important ideas presented in this work. However, it poses several intriguing and testable hypotheses, which, if tested could dramatically add to the impact of the work. For example, the separation of function mutations in SC components that promote crossovers but fail to polymerase and prevent the presumed increase in stiffness; this second 'wave' might also be eliminated by a time course analysis and/or conditional depletion.

4. The conclusion that the effect of Pch2 occur 'throughout pachytene' should be softened. They are based on an artificial over-expression that occurs over a non-physiological time-scale (3 hrs) in a mutant scenario (*ndt80Δ*). While intriguing, it

does not establish that the modulation of the axis by Pch2 occurs throughout physiological pachytene.

5. Several key aspects of the Fourier transformation at the base of almost all conclusions in the test remain opaque and are quite circular. See specific points:

- Doesn't FFT analysis of periodic patterns expected to yield peaks at both the basic (i.e., lowest) repeated unit (in this case, 0.45 μ m) and all multiplications of this value (0.9, 1.35, etc.)? Along these same lines: please rephrase the unsubstantiated statement in line 121, "implying a mechanistic relationship".

- Given the nature of FFT analysis, isn't the 'reconstitution' discussed in lines 130-135 obvious? FFT is, in essence, a conservative transformation that can be reversed. In addition, how is the correspondence tested statistically (in other words, what would be the null hypothesis an inability to reconstitute the pattern)?

- A bigger issue with the FFT analysis is the lack of a control dataset that lacks periodicity, or aspects of it. Given the nature of the correlations discussed here, is there a way to generate a non-correlated dataset that would retain some of the salient features in the empirical data (e.g., amplitudes, intensity distributions), but would shuffle peaks between different chromosomes? That would go a long way into demonstrating the robustness of the analysis, and would also allow the authors to conduct statistical analysis, which is currently mostly lacking.

Other points:

Line 76: foci of Zip3 (and other ZMMs) represent sites of future (or designated) crossovers.

Line 122: What does "statistically robust" mean?

Lines 137-138: Isn't the narrowness circular, due to selecting periodicities <0.6 μ m?

Line 303 and Supp Fig. 5: There is no statistical test for the better fit of the 'two-tiered' simulation versus the 'sprinkled' simulation, or for the "accurately recapitulate" claim in the main text.

(Remarks on code availability)

Reviewer #3

(Remarks to the Author)

This is an interesting paper that uses a detailed mathematical analysis of fluorescence intensity patterns for three proteins (Zip3, Hop1, Zip1) in yeast during prophase I of meiosis to define their distribution along chromosomes. They identify two patterns co-occurring on the chromosomes – a shorter periodicity “focal” pattern that is especially prominent for Zip3, but also observed for the other two, and a longer periodicity “domanial” pattern. They show that the domains (which are defined primarily by Hop1 and Zip1) have peaks that also show characteristics of interference. Both focus and domain periodicities seem to shorten in pch2 and top2 mutants, which were previously known to affect interference. They also make the case that because the relative occurrence of the foci vs domains is the same as the relative occurrence of “canonical” versus “non-canonical” crossovers, that the latter correspond to the domain peaks. The paper is dense, and in places took several reads to follow, but overall, I think the arguments are sound. In particular, the results and interpretation raise important hypotheses for the field. Whether they end up being completely or only partly right is less important than that it will trigger a new direction of research to test the hypotheses raised – and that makes this an important contribution!

I have a few questions/comments/suggestions.

The “domains” confused me a bit, since there are clearly two distinct types lumped together. From the graphs shown, it is very clear that although the three types of molecules do indeed co-occur, there is more to the story than the mathematical treatment really captures. The authors do acknowledge that enrichment varies, but don't really deal with it. Most strikingly, for domains, Zip1 and Hop1 anti-correlate – high Zip1 domains have comparatively less Hop1 and vice versa. This is consistent with the literature, but to me raises the question - can these different “domains” really be treated as the same sort of peak? Does it not mean something?

Moreover, it seems that in the examples shown, the Zip3 focal pattern might be more pronounced, and maybe have a tighter periodicity in the Hop1-rich regions than in the Zip1 rich regions. Is this real? Does it make sense to do analyses of these regions separately, if even just to make sure it's OK to lump them together?

By not going into detail on the (very interesting!) patterns observed with pch2 mutants, I think there is some opportunity being missed. For example: (a) is it real that the foci of Zip3 are largely lost in the mutant as it seems from Fig5A? If so, this is important, no? (b) It also seems important to note that while Zip1 and Hop1 enriched domains seem to anticorrelate in WT, in pch2 they overlap perfectly. I know this is discussed to some extent, but it also seems there is more to this story... (c) It seems adding back PCH2 on the CUP promoter seems to make the focal pattern even more intense and (it seems) the domain pattern less pronounced for all the molecules than it is in WT (Fig 5H). Does PCH2 play a bigger role in patterning or at least making the “foci” more prominent? Does this role explain how it plays out for “domains”? Is PCH2 in the rescue lines expressed at a higher level than in WT? Overall - Just saying that mutation of Pch2 does not alter the fact of the pattern, just relative amounts of the three types of molecules (as the paper currently does), seems to gloss over and over-simplify a lot of interesting and potentially informative/important patterns. It also glosses over the fact that the foci and domains may be differently affected, and that there is clearly something intriguing going on with the Zip1/Hop1 domains. What happens to crossovers in these lines, by the way?

I agree that the similarity of patterns of domains and “non-canonical crossovers” is highly suggestive that domain peaks correlate with these. Are the Zip1-enriched domains and Hop1-enriched domains equal in this regard? Can these “non-

canonical" sites be cytologically visualized somehow? Since the domains cover most of the chromosome, of course they will correlate, but can one map where exactly? Are they occurring in the middles of domains? Maybe there is no marker for them... Just curious if there is a clear way to functionally link them.

Fig 3C – the overlain traces and the labels are very hard to distinguish. Please make the colors more distinct. The Black/grey in the Zip3 panel works, but the labels are both grey. For the rest of the traces, it would be helpful to make them clearer, and the labels too.

Fig 3B – would it maybe make sense to add a replicate of the WT panel from Fig 2 so it's easier to compare?

Line 119; "The same" seems a strong statement – maybe add "about" or say "similar"? I can see that 0.41 and 0.42 are arguably the same. But is 0.8 really the same as 0.96?

Overall, I have a feeling they are right, and I think their discovery is an important one – that there might be two phases of interfering crossovers and that the two types might not be as different as previously thought is an insight that will be of great interest in the field and will stimulate follow-up research. I would just like to see more discussion of some of the patterns observed. Especially important to me are the PCH2 vs pch2 patterns and the meaning (or lack thereof) of Zip1-enriched vs Hop1-enriched domains.

(Remarks on code availability)

Version 1:

Reviewer comments:

Reviewer #1

(Remarks to the Author)

I am happy that the authors were ready to implement almost all of my suggestions and argued convincingly in the remaining cases. I have no further concerns or suggestions and recommend publication of this challenging, but important study.

(Remarks on code availability)

Reviewer #2

(Remarks to the Author)

The revised manuscript by Kleckner, White and colleagues did not address the major concerns raised by me and by other reviewers concerning the original submission. My assessment of the manuscript therefore remains the same. The descriptive results are indeed intriguing. However, many of the mechanistic claims are not supported by the data, and there are no sufficient controls to rule out experimental artefacts. A speculative hypothesis raised by this work - that "domainal triads mark the sites of a previously mysterious minority set of crossovers" - is presented in the abstract as an undisputed result. The limited resolution of the raw data remains a limitation of the experiments and a persistent source of doubt, and is further confounded by multiple additional layers of over-analysis and over-interpretation.

(Remarks on code availability)

Reviewer #4

(Remarks to the Author)

This review focuses on evaluating on the Fourier and imaging analysis as requested and the points made by the prior reviewers. The manuscript centers on the validity of applying these techniques properly.

Fourier analysis is a good way to obtain the various dominant frequency components within an image. However, typically it is most robust if there is a strong repetitive component within the image. However, it can artifactually pick up components that can be misinterpreted as having a certain periodicity. Zip3 and Zip2 foci in yeast have been shown to have a weak interfering distribution along meiotic chromosomes and using Fourier analysis as shown in this manuscript, periodicities were found. Some questions that have not been addressed satisfactorily is accuracy of the periodicities and their relationships among the 3 sets of proteins (Zip2/Zip3, Hop1 and Zip1) for the following reasons:

1) There was a range of chromosome sizes used and chromosomes were not identified in these spreads. The density of the chromosomes of the Zip3 foci at smaller sizes are different than those of the larger sizes. Given the way the periodicities were identified, it seems that the identified frequencies would be changed depending on the chromosomes used. Also edge effects would come into play given the low number of foci per chromosome. An experiment that would address this is to divide the spreads into small and large chromosomes and see if there is the same periodicities obtained.

2) Zip2 and Zip3 foci don't overlap exactly, but it is assumed that they do and thus can be used interchangeably. In the literature, Zip2 foci seem to be significantly fewer than Zip3 foci. It thus is not clear one can combine the data from both.

3) The distribution of Zip1 and Hop1 are much less punctate than Zip2 and Zip3 foci, thus you would always pick up some periodicity in Zip1's and Hop1's signal that would match periodicities that would be found in Zip2/Zip3 as well as potentially other periodicities. Reviewer 2 has a point that some of the focal peaks in Hop1 and Zip1 are minor which is part of the worry that both Zip1 and Hop1 are distributed throughout the chromosomes and that one can always find periodicities that will match Zip1/Hop1. An experiment that would make these results more convincing is to show that this same distribution of focal peaks does not occur using a protein that is not associated with crossovers but is also found along the axis.

4) Reviewer 2 raises an important point that the authors did not use super-resolution microscopy which would better define actual focal positions. There is a strong possibility that the authors may get entirely different results if they could actually distinguish foci that are close together as distinct foci. At their resolution these foci would be merged and if one does not have an accurate depiction of the foci, it makes the analysis less convincing. An experiment that does not require reacquiring imaging data from the spreads is just to show that in the majority of cases that the distribution of foci using super-resolution is not significantly different than that obtained using conventional fluorescence microscopy for one identified chromosome. It would require marking a specific chromosome but that is relatively easy in yeast.

Overall, I agree with how Reviewer 3 phrased his last comment in that he/she feels they are right. "Overall, I have a feeling they are right, and I think their discovery is an important one – that there might be two phases of interfering crossovers and that the two types might not be as different as previously thought is an insight that will be of great interest in the field and will stimulate follow-up research." I also think that the data feels like it should be right but I do not think the authors have rigorously shown that it is right. Thus at a low resolution, it is very correlative but not definitive.

(Remarks on code availability)

Version 2:

Reviewer comments:

Reviewer #5

(Remarks to the Author)

My specific comments regarding the issues raised by Refs 2 and 4, and of other issues in the manuscript are described below.

I am not sure I understand how the bleedthrough analysis in Supp Data 1 really proves there is no bleedthrough. This should have been done in separate experiments where only ONE of the proteins is labeled and the three channels are acquired.

While length analysis is consistent with their traces corresponding to chromosome XV, it is unclear why the authors did not label this chromosome directly, i.e. by using oligopaints. As other chromosomes have similar lengths (VII, XII), this puts into question of whether they are really looking at chr XV. However, this should not be an issue if the frequencies of the spots are in um as argued by the authors.

The nature of the signal (diffraction limited images) is such that it is difficult to visualize directly the structures the authors are aiming to detect. A convincing validation of whether their algorithm is able to deconvolve multiple overlapping signals is absent.

On point 4.2, ref 4 argues that Zip2 and Zip3 should not be used interchangeably. The data (Supp Data 4D) provided by the authors is not convincing in supporting the authors' statement that these two factors can be interchangeably used, as there are evident changes in the spatial distributions shown. Therefore, while the distributions of these factors seem to be very similar, they are not identical.

I am not convinced of the Fourier analysis used. Typically in such analysis one expects to find directly peaks in the Fourier space, which they don't seem to find (e.g. Supp Data 3B). Instead they seem to rely on derivative analysis to identify changes in the slope of the Fourier function. I have never seen this analysis in the past and they do not show solid evidence of why it has to be used or that it works.

The authors argue that spatial resolution is not an issue. However, I would side with the reviewers' arguments as the resolution of the images shown (e.g. Fig 1) shows ill defined spots, with intermediate densities within spots that could actually arise from additional clusters not resolved by conventional microscopies. While SIM doubles the resolution of these microscopies, the increase in resolution is often not sufficient. I would highly recommend the authors to use either STED or STORM, as these would afford a 5x or 10x increase in resolution that would settle the issue.

I don't agree with the authors' argument that SIM or other methods cannot be used with their existing samples as SIM/STED/STORM could be applied using immunofluorescence as the authors use for conventional imaging.

I am even confused by the rebuttal of point 4a, as the authors talk about periodicities of 0.23 um, while in previous paragraphs they argued their conventional microscope is able to detect the spatial variations in the sample. However, 0.23 um is at the resolution limit of the microscopy methods they used.

Finally, it is hard to assess the responses to the issues raised by Ref 2, as I have no access to previous interactions with the original issues raised by Ref 2. This said, I agree that there are issues regarding reliability of their Fourier method to identify peaks, of whether there is bleed through, whether Zip2/3 can be used interchangeably, and finally of whether the spatial resolution is enough to properly support their interpretation and model.

(Remarks on code availability)

Version 3:

Reviewer comments:

Reviewer #5

(Remarks to the Author)

The authors have provided plausible explanations to most of my concerns.

(Remarks on code availability)

General Comments

We appreciate all of the comments and suggestions of the reviewers. Taking them into account has resulted in a significant improvement to the manuscript. All basic conclusions remain unchanged and several insights have been significantly sharpened.

The most important improvement has been our realization, triggered by a comment from Reviewer 1, that there are four prominent periodicities in quantitative intensity profiles (rather than three as in our previous analysis). We also note that the existence of these four periodicities is now validated not only by FFT analysis but by a second, independent method (frequency comb) which is not subject to the same potential complexities as FFT. Furthermore, we can show that the shorter two of these periodicities, and their relative prominences, are also characteristic of canonical crossovers and, in fact, are an expected consequence of the way that crossover patterning works. In brief: undifferentiated precursor recombination complexes are evenly spaced. As a result, inter-crossover distances are quantized. Given the known distance over which crossover interference acts in yeast, exactly the two observed preferred spacings/periodicities, and their relative prominences, are predicted according to whether adjacent events are separated by two “inter-precursor distances” (i.e. one intervening unreacted precursor) or, more rarely, two “inter-precursor distances” (i.e. two intervening unreacted precursors). The same effect can analogously explain the two periodicities of domainal triads, taking into account their longer interference distance.

On a practical level, the presence of these four periodicities has allowed us to reconstruct the individual intensity profiles for focal and domainal signals by cleanly dividing FFT frequencies into two groups using a single threshold that separates the two shorter and two longer periodicities, with no frequencies omitted. (Our original analysis effectively excluded the minority periodicity of focal/canonical crossover events). The resultant reconstructions very closely report the well-defined positions of focal signals and the clear shapes of domainal signals in experimental data, even more so than in our original analysis. They also now allow to show that, on a per-chromosome basis, the peaks of focal and domainal signals are interdigitated. This feature, supported also by other data, has interesting functional implications and provides direct evidence that the two groups of FFT frequencies are reporting different features.

Additional comments of Reviewers 1 and 2 regarding the analysis are addressed below.

We have also addressed the desire of Reviewer 3 for more information regarding the nature of Pch2 roles. Our important general conclusion that Pch2 acts specifically on domainal triads, with two roles which, together, can explain previously-described cytological loading patterns.

Finally, we have restated the evidence and arguments for our conclusion that the two tiers of triads correspond to canonical and minority crossovers, respectively. The case is very strong, and this presentation should alleviate reviewer hesitations.

Reviewer 1

White et al. describe an analysis of immuno-labeled chromosome spreads of yeast nuclei, primarily, but not exclusively, in the *ndt80Δ* arrest. They trace the label along chromosome midlines and find that the observed patterns contain 2-3 periodic components. By separating short from long wavelengths through Fourier analysis, they find that the short wavelengths largely fit what is in cytology referred to as 'foci', while the longer wavelengths correspond to what was described as 'domains' of the corresponding proteins. Strikingly, both foci and domain patterns co-localize among three key components of meiotic chromosomes: Hop1, Zip1, and Zip3. Foci patterns arguably correspond to majority COs, since they align with Zip3, while the domain patterns fit those of 'minority' COs.

In general the paper is well written and performed with great insight. The approach to trace chromosome signal and employ a periodicity detector such as Fourier-analysis is interesting. It is challenging to critically understand the work. I think that the analysis shows important points. My comments below aim to clarify issues and to help to engage the readership in critically retracing reasoning and results.

We really appreciate this reviewer's appreciation of our work.

i) Tracing chromosomes - pixelation. Yeast chromosomes are small, even in pachytene. When analyzing or measuring linear features in images, especially when using a limited number of 3 pixels for the width of the tracing, the shape and orientation of the object can significantly affect the measurement. The authors are very likely aware of the problems. It would nevertheless be helpful to address these problems briefly, so as to keep the audience aware.

I have illustrated some obvious pixelation issues in a sketch below.

a) Changing the relative orientation of the same object relative to the pixel grid will affect its representation.

The object is perfectly aligned with the pixel grid in A - but rotated by 45° in B. Below the object, I plotted how the object is likely represented after pixelation - intensities are now distributed over a larger set of pixels with reduced intensities. Thus: Straight objects can be represented differently depending on their alignment with the pixel grid.

b) In addition, the 'midline' (14 pixels in A) is represented by only 10 pixels in B, due to their diagonal arrangement.

The question arises, how a "width of 3 pixels" is realized under the circumstance of a 45° rotation. It's difficult because, due to the reduced number of pixels in the midline, the number of adjacent pixels may also be reduced, when at the same time the label is smeared over more than the original 28 pixels.

c) kinks and bends

Objects, which are bent or have kinks of various degrees, may compress or decompress label accordingly, possibly causing fluctuations in what might otherwise be a flat intensity in the straightened state (C). c) The problems may be aggravated when chromosomes with kinks or bends (C) are arranged arbitrarily relative to the grid.

Suggestion: It would be great to comment in the text, why these issues do not affect the Fourier outcomes. In the supplement or computational methods section, one might explain how the algorithm

deals with pixelation. If one had a graphical representation, that would, of course, be of the highest didactic value.

Responses to pixelation issues discussed above. In brief: we have used a standard method which does not have the limitations that concern the reviewer, and we now cite this approach in the Methods and summarize it, with control illustrations, in Supplementary Fig. 2.

Specifically: we traced chromosome paths using the segmented line selection tool of open-source software FIJI, and then extracted the corresponding pixel intensities using the software's Plot Profile command. This is the standard approach for such analysis in the field. We have used it ourselves for previous analyses of Zip3 foci along pachytene chromosomes. This algorithm first takes the manually defined trace line and computationally straightens the corresponding chromosome image such that the trace line is now straight. The signal intensity at each position along the straightened chromosome is then determined, for the specified line width across the chromosome at that position. Given that signal intensities are relatively symmetrically distributed radially along the length of the chromosome, with higher intensities centrally than peripherally, it can be expected that the same relative intensities will be obtained irrespective of line width, as long as that width is not so wide as to include many "empty" pixels. We selected a line width of 3 pixels to include most of the signal without including empty pixels. In fact, reconstruction experiments show that line widths of 1, 3, and 5 pixels give indistinguishable intensity profiles, as expected (Supplementary Fig. 2B).

ii) Tracing chromosomes- selection

Based on the strain table, it is possible that specific chromosomes (II and XV) were selected for tracing. If there was any selection, it would be beneficial to add this description to Figure 1. Given that three channels were used for the three proteins, it is unclear whether a channel was available to recognize a chromosome. If chromosomes of different lengths were traced, were always all chromosomes of a spread traced, or only those that appeared continuous? Or were chromosomes with a minimal length selected?

If chromosome lengths varied, a comment on the effect of the length distribution on the resulting Fourier coefficients would be highly appreciated. In cases where specific chromosomes were analyzed, it would be beneficial to announce this prominently (e.g., Fig. 1).

We did indeed fail to specify how chromosomes were chosen for analysis. To answer the reviewer directly: specific chromosomes were not selected. Instead, we selected chromosomes that were of about 3 μ m in length. We have now made these points in the text, with an accompanying histogram showing the lengths of the chromosomes analyzed, which are distributed around an average of \sim 3 μ m (Supplementary Fig. 3A).

ii) Fourier analysis - PSD

or readers who are not very familiar with Fourier analysis, the magnitude plot may be more accessible than the power spectrum density (PSD). PSD is essentially the square of the magnitude and is often used when the result is interpreted as the "energy" of the underlying frequencies. The coefficients of magnitude could be interpreted as relative contributions of the respective frequencies to the signal, which is fairly intuitive. Of course, the authors should use what they think is most appropriate.

We agree. We now use magnitude rather than PSD to display the FFT output (Fig. 2A).

ii) Fourier analysis - arguments/binning

The PSD plots are presented as semi-log plots, where the arguments are plotted linearly and the PSD coefficients in log scale. However, it appears that for the arguments bins of exponentially decreasing size were used, resulting in a very crowded lower left part. It would be beneficial if the complete list of arguments could be presented (such as the law that generates them) together with the rationale for choosing this particular set.

If a magnitude plot were presented in a log-log format, this would decluster the left corner and the reader might be able to interpret the coefficients as relative contributions of frequencies to the signal. In its current form, it is very difficult to interpret the values due to the squaring and the clustering of values in the left part.

We now plot the arguments on a linear scale, i.e. in units of frequency (1/μm) rather than periodicity (μm), Fig. 2A. Since we no longer plot the PSD (above), we now also plot the y axis on a linear scale. This makes the primary output much clearer in the range of frequencies/periodicities that are of interest to the current study.

ii) Fourier analysis - two 'most prominent' peaks

It might be helpful for the reader if the criterion for selecting the two 'most prominent' peaks in the PSD were stated in the figure legend or text close to its first occurrence. For instance, stating "the two local maxima with the longest periodicities were chosen" would be consistent with the six peaks shown in Fig. 2A. In the absence of thresholding, this step produces unstable values, which the authors compensate for with statistical analysis over the complete dataset.

We have included language which states that we identified, in each FFT output, the periodicities with the two highest amplitudes in the frequency range of interest (i.e. 0.5-5/μm).

Line 130: "The most important FFT periodicities for each profile, in the frequency range of interest (0.5 – 5/μm), are given by the two most prominent FFT signals, i.e. the peaks (local maxima) of highest amplitude in that range (asterisks; Fig. 2A)."

ii) Fourier analysis - statistics of power density peaks

The $n \sim 2$ peak periodicities (per protein and chromosome) are approximated by a combination of three Gaussian curves. My feeling is that the densities are noisy, especially for the longer frequencies, potentially signaling that the collected data are on the lower limit. For example, for the important Zip3 density, it is not obvious that it is best approximated by 'three Gaussians'. While the analysis of the data, as it currently stands, yields reasonably robust results after fitting, it would be fantastic to actually see the three Gaussian peaks emerge also for Zip3, which is currently too noisy to show three peaks. To make sure the "three Gaussian" model is really justified, Fig. 2 might benefit from a 2-3 fold increase in the data set.

If the three peaks show clearly for each protein in one data set (i.e., Fig. 2), the other data sets could be approximated in the same way— even with more sparse data.

As discussed above, we have now appreciated that the FFT output is best described by a set of four peaks, two sets of major and minor peaks, with the two sets corresponding to focal and domain signals, respectively. The four-peak fits are presented in Fig. 2B. These four peaks are identifiable in the averaged FFT output of all 81 measured chromosomes (Supplementary Fig. 3B). They are also validated by frequency comb analysis, which defines the same four

peaks by an entirely unrelated method, as presented in Supplementary Fig. 3C. We understand that the data seem noisy, but the conclusions are robust, not only because of frequency comb validation, but also because of their match to previous data (for Zip3 focus patterns) and their coherent and synthetic explanation of all experimental data.

ii) Fourier analysis - inverse FFT

In this analysis, frequencies $< 600\text{nm}$ and frequencies $> 900\text{nm}$ were used to define high and low frequency components of the signal. Strikingly, for this example, the high frequency patterns of all three proteins show convincing colocalization. On this occasion, the authors may comment that they checked for 'bleed through' in their three channels, which could also lead to striking colocalizations.

We were remiss in not presenting the relevant control data. Two lines of evidence against bleed-through are now cited in the text, for one of which, validation is in Supplementary Fig. 1.

For orientation, it would be beneficial to plot the sum of the two inverse FFTs ($<600, >900\text{ nm}$) as a dashed line in the top panel of Fig. 2C, to demonstrate how well the two frequency ranges capture the original profile and confirm that nothing important important was discarded.

This was an extremely helpful comment. It triggered our appreciation that, essentially, the "missing frequencies" correspond to a fourth peak, with the implications outlined above. Among diverse other advantages, we can now divide the FFT frequency outputs cleanly into two groups, using a single threshold, with nothing missing. The corresponding reconstructions very closely match experimental data, as shown in Fig. 2C, and permit a coherent synthetic explanation for the observed periodicities.

ii) Colocalization of high frequency peaks ('foci')

These data are quite clear. Please indicate somewhere prominent, e.g. in the legend, that this statistics is for the complete data set and not only for the example chromosome.

Yes. This is now clearly stated in the legend for Fig. 2.

Line 1014: "(G and H) Distributions of distances between each Zip3 peak and its closest Hop1 peak (left) and analogous distributions for Zip3 – Zip1 and Hop1 – Zip1, for focal and domainal signal components of all measured chromosomes (n = 81 chromosomes)."

iii) Coc

Coc: It may be helpful to repeat the Coc formula (as in Zhang, 2104, Fig. 1 c) in the figure, mentioning in the legend that chromosomes were split into 30 (foci) or 15 (domain) equal intervals.

Yes. Such a description is now presented in Fig. 3B with a brief summary in the main text.

The other results and the discussion follow smoothly from the discussed parts.

Reviewer #2 (Remarks to the Author):

The manuscript by White, Kleckner and colleagues carefully examines the distribution of three

key factors along meiotic chromosomes in order to understand the mechanism underlying the distribution of crossovers. This is an important and timely question. The patterning of crossovers has implications for a wide array of fields, from basic chromosome biology to infertility to evolution. Crossovers are uniquely positioned in every meiocyte; however, they exhibit intriguing patterns of co-occurrence, both at the population level and relative to one another in every single meiosis. This work is timely since recent work on condensates led to the reexamination of several of our assumptions about the mechanisms that pattern crossovers. In addition, advances in microscopy techniques and genetic analysis in multiple organisms allowed us to gain unprecedented knowledge of meiotic chromosomes.

The authors revisit a cytological approach to localize meiotic structure on budding yeast meiotic chromosomes. This is an appropriate method to study a phenomenon that is 'single-molecule' in nature. They develop an automated analysis pipeline using minimally processed images to reveal difficult-to-discern patterns of distribution. They confirm this approach by recapitulating previous observations of the regularity of crossover positioning but also reveal previously undocumented 'domainal' periodicities. This analysis leads to the conclusion that a large-scale periodicity reflects 'minority' crossovers (also called non-interfering or class II crossovers). These events, which represent a minority of exchanges, are poorly understood, so defining a cytological correlate for them would be of high significance. They then integrate their findings to model a two-step patterning process that can recreate empirical distribution of crossovers.

The conclusions of this work, if substantiated, would be of high interest to the meiosis and chromosome biology communities. Given budding yeast's centrality as a model organism for meiosis, this work would be of wide interest to researchers working in all model organisms. However, multiple potential issues related to image acquisition, mathematical analysis, statistical testing and lack of empirical testing of the proposed hypotheses cast doubt on the conclusions.

We respectfully disagree strongly with this assessment. We address the specific concerns of this reviewer below.

Below are some of the most crucial issues:

1. As the authors themselves claim, the 'fundamental findings of this study' rely on fluorescent micrographs. The authors test two potential artifacts in these images - two different proteins that localize to future exchanges (Zip2 and Zip3) and both physiological and arrested (*ndt80Δ*) meiocytes. However, a few other controls are essential.

First, the observed effects are on a similar size scale as both the pixel size (67nm; 3-pixel averaging brings it to close to **half of the observed focal periodicity**, and even more than half of the focal periodicity in some mutant scenarios; it also happens to be the hypothesized spacing between precursors) and the diffraction limit of fluorescence microscopy (~150-200nm). It is essential to confirm that these technical aspects do not influence the analysis. Some of the possible solutions are to acquire and analyze similar images using a super-resolution microscopy system with different resolution and pixel size, and to change pixel size in this system (e.g., by adding an optovar).

*In principle (Nyquist theory), the pixel size used in this study (67 nm) allows us to resolve features that are separated by 134 nm and above. By smoothing the reconstituted 'focal component' intensity profiles using a 3-pixel moving average, we removed the ability to resolve features that are closer than 201 nm (67 nm * 3) apart. This makes sense because, given the resolution of light (~200 nm), any such features in our data are likely to be noise.*

We find that focal peaks are separated, on average, by ~450 nm, well above the resolution limits of our experimental set up. Furthermore, as discussed above, the focal peaks defined here by FFT-reconstruction and smoothing, very closely match Zip2/3 foci identified by orthogonal approaches that do not involve smoothing (Fig. 3). Thus, definition of this periodicity for other molecules, as well as the definition of longer periodicities for all molecules, are all valid.

It is of course theoretically possible that future work, e.g. by single molecule localization microscopy, will resolve additional features of interest whose spacing is below the diffraction limit of light. However, for the conclusions of the current work, we have no evidence that such an undertaking is either necessary or useful. Correspondingly, if such features were to be defined, they would not invalidate any of the findings of this study.

Second, the minor amplitude of some of the effects (e.g., the 'focal' peaks in the Hop1 & Zip1 data), raises worries about bleed-through between the fluorescent channels. This should be tested by swapping secondary antibodies and repeating the analysis.

This issue was also raised by Reviewer 1 and we apologize for omitting the necessary controls in our original submission. We now show that spectral bleed-through is not a problem (Supplementary Fig. 1). We also clarify in the text that results were unaffected when a different combination of secondary antibodies was used.

Line 199: "In addition, Zip2 data were obtained using a different combination of secondary antibodies from those used for Zip3 analyses above (Methods), further indicating that the observed patterns are not an artefact of spectral bleed-through (above)."

A final suggestion is not a requirement but would be of interest. Advances in microscopy should enable analysis of chromosomes in structurally preserved samples, rather than the chromosome spreads used here. Given the large body of work from the Kleckner lab and others on this system, it would be beneficial to confirm the existence of these patterns in multiple sample preparation protocols, ideally including minimally perturbed ones.

This is a lovely suggestion. However, in our view, the results presented strongly support the basic conclusions of this paper without applying any entirely new, unvalidated and currently unavailable method. (We also note that analysis of individual yeast chromosome paths in whole cells is impossible.)

2. The conclusions relating to the correspondence of domainal triads to class II crossovers are premature. They are based on mere correlations, and are not tested. Without a way to substantiate these correlations, this conclusion should be moved to the Discussion and clearly be marked as speculative. (See point 3, however.)

We respectfully disagree. Our original presentation of the reasoning and evidence regarding minority crossovers was not adequate. The evidence for our conclusion is, to us, compelling and overwhelming. We have rewritten the relevant section of the Results and accompanying Discussion section to better present the evidence.

We also note that use of the term "Class II crossovers" needs to be modified. It has been used indiscriminately in the field to refer both the minority crossovers that occur in wild type meiosis and the many crossovers that arise in certain mutant situations, which are likely to be something else entirely.

3. The modeling of crossovers in the beam-film framework as two waves with different

properties is intriguing. By itself, its 'economic' nature does not significantly add to, or validate, any of the important ideas presented in this work. However, it poses several intriguing and testable hypotheses, which, if tested could dramatically add to the impact of the work. For example, the separation of function mutations in SC components that promote crossovers but fail to polymerase and prevent the presumed increase in stiffness; this second 'wave' might also be eliminated by a time course analysis and/or conditional depletion.

We are glad that the reviewer is sufficiently intrigued by our model to envision further experiments to test its validity, all of which would be interesting to pursue. However, in our view, the suggested experiments are a task for the future, even aside the fact that they are all substantially non-trivial.

4. The conclusion that the effect of Pch2 occur 'throughout pachytene' should be softened. They are based on an artificial over-expression that occurs over a non-physiological time-scale (3 hrs) in a mutant scenario (ndt80Δ). While intriguing, it does not establish that the modulation of the axis by Pch2 occurs throughout physiological pachytene.

We agree and have modulated the language now to read as follows:

Line 449: "Thus, in wild type meiosis, Pch2 may have the potential to modulate the levels of triad components continuously throughout pachytene without affecting the existence of basic triad structure per se."

5. Several key aspects of the Fourier transformation at the base of almost all conclusions in the test remain opaque and are quite circular.

We respectfully disagree with these characterizations. Some answers to these concerns are contained in our general discussion above. In addition:

See specific points:

- Doesn't FFT analysis of periodic patterns expected to yield peaks at both the basic (i.e., lowest) repeated unit (in this case, 0.45um) and all multiplications of this value (0.9, 1.35, etc.)? *FFT analysis can indeed produce artifactual harmonic periodicities under some conditions. However, there are four lines of evidence that this is not the case in the present study. (i) The existence of shorter and longer periodicity signals is apparent in the primary experimental data and thus is not an artifact of subsequent analysis. (ii) We validate the existence of the (now four) primary component periodicities by a second approach, frequency comb analysis, which has an entirely different logic from FFT but nonetheless gives the same result, as described in Supplementary Fig. 3C. (iii) Reconstructed intensity profiles for the two defined groups demonstrate that peaks for the two groups are interdigitated (Fig. 2C, bottom), a conclusion that also emerges from other analyses presented later in the paper (Fig. 4B – E and Supplementary Fig. 6B and D). This situation is incompatible with harmonic effects, which would predict that peaks of different periodicities would occur at different subsets of the same positions. (iv) We show that Pch2 specifically affects domainal triads. This would not be the case if domainal triads were a harmonic artifact of focal triad patterns.*

Along these same lines: please rephrase the unsubstantiated statement in line 121, "implying a mechanistic relationship".

This statement has been eliminated, along with a general rewriting of this section of the paper.

- Given the nature of FFT analysis, isn't the 'reconstitution' discussed in lines 130-135 obvious? FFT is, in essence, a conservative transformation that can be reversed. In addition, how is the

correspondence tested statistically (in other words, what would be the null hypothesis an inability to reconstitute the pattern)?

Inverse FFT is, as the reviewer states, a simple reversal of an original FFT transformation. Of course, if you apply this to the entire set of FFT frequencies, you will get back the original data. However, this was not how we applied inverse FFT for this work. In our approach, we applied inverse FFT to two distinct subsets of periodicities according to a defined threshold (above). In this way, we could reconstruct the primary intensity profiles corresponding to those two subsets periodicities which, then, could be subjected to further analysis. We have clarified the language in the paper which describes this logic. In any case, there is no reason to apply statistical analysis because, as the reviewer says, inverse FFT is intrinsically a conservative transformation whose validity is implicit.

- A bigger issue with the FFT analysis is the lack of a control dataset that lacks periodicity, or aspects of it. Given the nature of the correlations discussed here, is there a way to generate a non-correlated dataset that would retain some of the salient features in the empirical data (e.g., amplitudes, intensity distributions), but would shuffle peaks between different chromosomes? That would go a long way into demonstrating the robustness of the analysis, and would also allow the authors to conduct statistical analysis, which is currently mostly lacking.

We disagree with this comment. We appreciate that the data seem somewhat noisy, but there is no ambiguity as to the results. There are now many convergent lines of evidence which validate our conclusions. The four groups of periodicities that we detect by peak analysis of FFT outputs of individual chromosomes (Fig. 2B) are detectable in the averaged FFT output of all 81 measured chromosomes (Supplementary Fig. 3B). They are also validated by frequency comb analysis, which defines the same four peaks by an entirely unrelated method, as presented in Supplementary Fig. 3C. In any case, for focal triads, presence of periodicity cannot be an issue. We know from many years of research that the signals are evenly spaced. Furthermore, simulation of random peaks will give random results, and this cannot, in principle, recapitulate the clear, synthetically related and biologically significant conclusions that emerge from our approach of defining prominent periodicities.

Other points:

Line 76: foci of Zip3 (and other ZMMs) represent sites of future (or designated) crossovers.

We think that the reviewer means line "96" of the submitted text, which read:

... "define peaks of Zip3 intensity ("foci") as sites of evenly-spaced crossovers"

The equivalent line in the revised text reads:

Line 89: "By this approach, Zip3 signals are defined as discrete "foci" that specifically mark the sites of "canonical" crossovers."

We appreciate the reviewer's point, but for us, marking the sites of crossovers does not specify that the crossovers have actually occurred, but rather that they are the sites of crossovers whenever they do occur.

Line 122: What does "statistically robust" mean?

This statement referred to the longest of the three periodicities originally defined by Gaussian fitting. This issue now goes away because we understand that there are four periodicities which now incorporate this longest one in a meaningful way.

Lines 137-138: Isn't the narrowness circular, due to selecting periodicities $< 0.6\mu\text{m}$?

This statement is based on the signals that appear in the reconstructions. It is clear from the original intensity profiles (i.e. raw data) that focal peaks are narrower than domainal peaks. The point of the reconstructions is that they accurately reflect the data. Thus, the properties revealed are intrinsic to the actual signals. An important general point is that we are not using FFT analysis to reveal features that are invisible - we are using it better analyze the features that are present in the data. In any case, the reviewer's statement is true for single sine waves but need not necessarily apply to sums of multiple sine waves.

Line 303 and Supp Fig. 5: There is no statistical test for the better fit of the 'two-tiered' simulation versus the 'sprinkled' simulation, or for the "accurately recapitulate" claim in the main text.

It is not useful to compare the "sprinkled" situation to the "two-tiered solution" because we have evidence for the latter, and there is no evidence for the former. We have therefore removed these simulations from the current version of the text.

Reviewer #3 (Remarks to the Author):

This is an interesting paper that uses a detailed mathematical analysis of fluorescence intensity patterns for three proteins (Zip3, Hop1, Zip1) in yeast during prophase I of meiosis to define their distribution along chromosomes. They identify two patterns co-occurring on the chromosomes – a shorter periodicity “focal” pattern that is especially prominent for Zip3, but also observed for the other two, and a longer periodicity “domainal” pattern. They show that the domains (which are defined primarily by Hop1 and Zip1) have peaks that also show characteristics of interference. Both focus and domain periodicities seem to shorten in pch2 and top2 mutants, which were previously known to affect interference. They also make the case that because the relative occurrence of the foci vs domains is the same as the relative occurrence of “canonical” versus “non-canonical” crossovers, that the latter correspond to the domain peaks. The paper is dense, and in places took several reads to follow, but overall, I think the arguments are sound. In particular, the results and interpretation raise important hypotheses for the field. Whether they end up being completely or only partly right is less important than that it will trigger a new direction of research to test the hypotheses raised – and that makes this an important contribution!

We appreciate the reviewer's enthusiasm. However, we note that, in our view, our findings are much more than "important hypotheses". As is now described in a clearer way in the new text, our results are important new findings that require the field to readjust its thinking after many years of wandering around in misconceptions and unwarranted and incorrect speculations.

I have a few questions/comments/suggestions.

The “domains” confused me a bit, since there are clearly two distinct types lumped together. From the graphs shown, it is very clear that although the three types of molecules do indeed co-occur, there is more to the story than the mathematical treatment really captures. The authors do acknowledge that enrichment varies, but don't really deal with it. Most strikingly, for domains, Zip1 and Hop1 anti-correlate – high Zip1 domains have comparatively less Hop1 and vice versa. This is consistent with the literature, but to me raises the question - can these different “domains” really be treated as the same sort of peak? Does it not mean something?

- The reviewer's confusion may stem from the fact that "domains" are not broad regions but, rather are defined as peaks of intensity, and that the triads are defined by colocalization of those peaks. We have tried to clarify the language that we use to describe this situation.

- Inspired by this comment, we have carried out additional comparisons of loading patterns in wild type and *pch2*Δ. We can now describe two specific roles of Pch2, show that they both pertain to domainal triads, and that these two roles can explain why cytological patterns are different in wild type and *pch2*Δ.

- We also note that the term "alternating hyperabundance domains" (for which we are responsible) could be interpreted to mean that Hop1 and Zip1 levels tend to be anti-correlated. We show that this is not the case. Rather, there are regions where the two levels are very different, sometimes giving the impression of anticorrelation (as in Figure 1), but on a population average level, this is not the case (see Supplementary Fig. 7D). This is now stated in the text:

Line 423: "We also note that the original description of Hop1 and Zip1 loading patterns referred to "alternating domainal hyperabundance" in wild type (and its abrogation in *pch2*Δ)¹¹. This description could be taken to mean that Hop1 and Zip1 loading tends to be anti-correlated. Quantitative analysis shows that this is not the case – the relative levels of the two molecules varies over a wide but symmetrical range, with no indication a bi-modal distribution (Supplementary Fig. 7D)."

Moreover, it seems that in the examples shown, the Zip3 focal pattern might be more pronounced,

and maybe have a tighter periodicity in the Hop1-rich regions than in the Zip1 rich regions. Is this real? Does it make sense to do analyses of these regions separately, if even just to make sure it's OK to lump them together?

The reviewer is not correct in this impression. As outlined above, there are not "Hop1-rich" and "Zip1-rich" regions, upon which "foci" are superimposed. Rather, previously defined "domains" comprise triads of colocalizing peaks of the two molecules (and Zip3). And while the heights of the peaks of different molecules vary along individual traces, focal triads (aka Zip2/3 foci) are, in contrast, evenly spaced all along the chromosomes.

By not going into detail on the (very interesting!) patterns observed with *pch2* mutants, I think there is some opportunity being missed. For example: (a) is it real that the foci of Zip3 are largely lost in the mutant as it seems from Fig5A? If so, this is important, no?

The reviewer is correct that Zip3 foci are less prominent in the mutant. This is because absence of Pch2 specifically increases the level of Zip3 (as well as of Hop1 and Zip1) in domainal triads (Fig. 5E and Supplementary Fig. 7B), which constitute most of the total signal intensity, and has no effect on the levels of any component in focal triads. This is now stated in the text.

Line 420: "(iii) Increased loading of components in domainal triads also explains the reduced prominence of focal triads, relative to domainal triads in *pch2*Δ."

(b) It also seems important to note that while Zip1 and Hop1 enriched domains seem to anticorrelate in WT, in *pch2* they overlap perfectly. I know this is discussed to some extent, but it also seems there is more to this story...

There are two issues here. The first is that Zip1 and Hop1 domains do NOT anticorrelate (above). The second is that, however, it is true that the levels of all three molecules do tend to covary along pch2Δ chromosomes, much more than along wild type chromosomes, which we have documented. However, this is true only in domainal triads. We have taken more care to specifically discuss the nature of this effect in the current text.

Line 369: "Pch2 promotes uncoupled loading of triad components specifically in domainal triads. Intensity profiles illustrate the fact that the levels of different triad components vary more coordinately along chromosomes in pch2Δ versus PCH2 (Fig. 5B versus Fig. 1C). This effect can be quantified. The normalized intensity level for each molecule was determined along each chromosome trace (e.g. Fig. 5C left). Differences in the levels of each pair of molecules were then determined, at every position along each individual chromosome, yielding a corresponding per-chromosome average (e.g. Fig. 5C right). The distribution of these average differences was then defined for all chromosomes in each population of interest for each of the three pairs of molecules. This analysis confirms that the relative levels of all three pairs of molecules are much more similar for pch2Δ chromosomes than for wild type chromosomes (Fig. 5D "Data", black versus grey bars; histograms in Supplementary Fig. 7A). The greatest effect is seen for Hop1 vs Zip1; an intermediate effect is seen for Hop1 vs Zip3; and the most modest, but still significant effect is seen for Zip1 vs Zip3.

The same analysis was then carried out for domainal and focal triad intensity profiles individually (Fig. 5D "Domain" and "Focus" and Supplementary Fig. 7A). Along wild type chromosomes, domainal and focal triads both exhibit substantial variation in relative levels of all three pairs of components, similar to those observed for total levels in primary experimental data. Furthermore, for focal triads, absence of Pch2 does not affect this variability for either Hop1/Zip3 or Zip1/Zip3, and only moderately reduces this variability for Hop1/Zip1. However, in striking contrast, domainal triads exhibit the same decreased differences in relative loading levels as the total (experimental) intensity levels for all pairs of triad components. Thus, Pch2 affects coordination of loading specifically in domainal triads.

Viewed in another way, these findings suggest that: (i) in domainal triads, in the absence of Pch2 activity, all three components exhibit coordinate fluctuations in their levels, implying that the loading of all three components tends to be functionally coupled. (ii) In wild type meiosis, Pch2 disrupts this coupling to give much greater variation in the relative levels of all three pairs of components in domainal triads. (iii) In contrast, Pch2 has no significant effect on relative component levels in focal triads, where loading of all pairs of components appears to be relatively uncoupled in both wild type and mutant situations, i.e. irrespective of the presence or absence of Pch2. Why domainal triads uniquely require Pch2 for uncoupled loading is an interesting question."

(c) It seems adding back PCH2 on the CUP promoter seems to make the focal pattern even more intense and (it seems) the domain pattern less pronounced for all the molecules than it is in WT (Fig 5H).

This could be true, since extended expression of Pch2 might lead to "over-removal" of all components from domainal triads, thus making focal triads more prominent. The analyses we have done thus far do not address this issue and/or do not reveal any obvious difference between wild type and "added back Pch2" cases (see Fig. 5).

Does PCH2 play a bigger role in patterning or at least making the "foci" more prominent? Does this role explain how it plays out for "domains"?

Well: we show that Pch2 specifically affects domainal triads in such a way that, in the absence of Pch2, focal triads are less prominent (see above).

Is PCH2 in the rescue lines expressed at a higher level than in WT?

Quite possibly, we have not measured the cellular concentration of Pch2 in these conditions.

Overall - Just saying that mutation of Pch2 does not alter the fact of the pattern, just relative amounts of the three types of molecules (as the paper currently does), seems to gloss over and over-simplify a lot of interesting and potentially informative/important patterns. It also glosses over the fact that the foci and domains may be differently affected, and that there is clearly something intriguing going on with the Zip1/Hop1 domains.

Yes - we agree and hope that we have now responded to this concern with further and clearer analysis.

What happens to crossovers in these lines, by the way?

*This has not been tested. These experiments were carried out in an *ndt80Δ* background where cells arrest at pachytene. This prevents genetic crossovers from being assayed. However, the original Boerner paper (Boerner et al. (2008). PNAS; 105: 3327 – 3332) demonstrated defects in *pch2Δ* in formation of crossover-specific intermediates and crossovers by DNA analysis, implying effects downstream of crossover/non-crossover differentiation.*

I agree that the similarity of patterns of domains and “non-canonical crossovers” is highly suggestive that domain peaks correlate with these. Are the Zip1-enriched domains and Hop1-enriched domains equal in this regard?

Again, there are no such things as Hop1- and Zip1-enriched domains (above).

Can these “non-canonical” sites be cytologically visualized somehow?

In our opinion, domainal triads ARE a cytological manifestation of these crossover sites. We have improved our presentation of the relevant findings, as discussed in the general comments above and Results and Discussion. In brief: the only evidence that Zip2/3 foci correspond to canonical crossovers is that they exhibit interference. These foci are sites of focal triads. Domainal triads are also a cytological feature that exhibits interference.

Since the domains cover most of the chromosome, of course they will correlate, but can one map where exactly? Are they occurring in the middles of domains? Maybe there is no marker for them... Just curious if there is a clear way to functionally link them.

The premise of this comment is not correct. Hop1, Zip1 and Zip3 always occur in peaks. Triads of domainal peaks are responsible for the cytologically-apparent domains. And the positions of these peaks define the positions of the domains.

Fig 3C – the overlain traces and the labels are very hard to distinguish. Please make the colors more distinct. The Black/grey in the Zip3 panel works, but the labels are both grey. For the rest of the traces, it would be helpful to make them clearer, and the labels too.

Yes, we have edited Fig. 3C and Fig. 3D to increase contrast between wild type and mutant data.

Fig 3B – would it maybe make sense to add a replicate of the WT panel from Fig 2 so it's easier to compare?

Fig. 3 has been revised and this data is no longer shown.

Line 119; “The same” seems a strong statement – maybe add “about” or say “similar”? I can see that 0.41 and 0.42 are arguably the same. But is 0.8 really the same as 0.96?

In response to reviewer 1, we have re-analyzed this data and found that the distributions are better described by four Gaussian distributions. The mean values of the component Gaussians

are now more consistent across all three molecules of interest (Zip3, Hop1 and Zip1; Fig. 2B), and this section of the paper has been rewritten.

Overall, I have a feeling they are right, and I think their discovery is an important one – that there might be two phases of interfering crossovers and that the two types might not be as different as previously thought is an insight that will be of great interest in the field and will stimulate follow-up research. I would just like to see more discussion of some of the patterns observed. Especially important to me are the PCH2 vs pch2 patterns and the meaning (or lack thereof) of Zip1-enriched vs Hop1-enriched domains.

We trust that we have now addressed these issues in adequate detail as described above.

REVIEWER COMMENTS

Reviewer #1 (Remarks to the Author):

I am happy that the authors were ready to implement almost all of my suggestions and argued convincingly in the remaining cases. I have no further concerns or suggestions and recommend publication of this challenging, but important study.

We gratefully acknowledge this reviewer's intuitive appreciation and understanding of this work as well as his/her substantive suggestions, which have greatly improved the paper.

Reviewer #2 (Remarks to the Author):

- The revised manuscript by Kleckner, White and colleagues did not address the major concerns raised by me and by other reviewers concerning the original submission. My assessment of the manuscript therefore remains the same. The descriptive results are indeed intriguing. However, many of the mechanistic claims are not supported by the data, and there are no sufficient controls to rule out experimental artefacts.

We respectfully disagree with the reviewer's assessment of our revised manuscript. We also find it difficult to respond to these reviewer comments.

1. In fact, we extensively and substantively addressed all of the previous concerns of this reviewer (and others). Among these, perhaps the most important is that this reviewer suggested that the identification of related periodicities might be an artifact of Fourier analysis. We addressed this concern specifically, at considerable expense of time and effort, by carrying out an entirely different, independent type of periodicity analysis, using "frequency combs" which revealed the same result as FFT analysis. We presented this analysis in the Supplemental Information. We have now pointed out the existence and significance of this analysis more prominently in the Results section

Line 150 - 155: "FFT analysis can, in some situations, artefactually define sets of related periodicities which do not actually exist. To exclude this possibility, we analyzed intensity profiles by a second, totally independent approach, frequency comb analysis. The mathematical basis of this approach is unrelated to that of FFT and thus is not subject to the same artifacts (Methods). Frequency comb analysis yields four periodicities that are very similar to those defined by FFT (Supplementary Fig. 3C)."

2. We also carefully addressed every one of the reviewer's other concerns regarding possible artifacts. Among these, we added additional controls showing that triads are not an artefact of spectral bleed-through (Supplementary Fig. 1).

We note that, in contrast, this reviewer has not responded to any of the revisions we made in any way, i.e. to indicate why he/she does not find them acceptable. Thus, this reviewer's comments do not present any actionable criticisms.

- A speculative hypothesis raised by this work - that "domainal triads mark the sites of a previously mysterious minority set of crossovers" - is presented in the abstract as an undisputed result. *In accord with this comment, and that of other reviewers, we have softened our conclusion in this regard.*

The Abstract now reads: "Diverse findings suggest that domainal triads mark the sites of a previously mysterious minority set of crossovers and, thus, that the two tiers account for all crossovers."

The title of the relevant section of the Results now reads:

Line 295: "Domainal triads likely mark the sites of previously mysterious "minority" crossovers."

The title of the relevant section of the Discussion now reads:

Line 568: "Focal and domainal triads likely account for all crossovers in wild type meiosis."

- The limited resolution of the raw data remains a limitation of the experiments and a persistent source of doubt, and is further confounded by multiple additional layers of over-analysis and over-interpretation.

1. We disagree with the comment that the resolution of the raw data is a limitation. As we pointed out in our previous response to Reviewer 2, and now discuss again below in response to Reviewer 4, there is more than sufficient resolution in our system to detect the four main periodicities in the primary data (~0.4, ~0.65, 1 and ~1.6 μm) and resolve adjacent focal peaks in the focal component (~500 nm spacing) and adjacent domainal peaks in the domainal component (~1 μm spacing). We have now stated this explicitly in the text:

Line 143 - 145: "All identified periodicities are above the resolution limit of our imaging system (190 – 240 nm; Abbe's diffraction resolution limit of light)."

2. We now also present, in the Supplemental Information, our analysis of a previous study of Zip1 localization using Structured Illumination Microscopy, which provides higher resolution. The results of that study directly support the specific conclusions that we draw in our paper, as follows:

Line 365 - 370: "Interestingly, a previous published study has imaged Zip1-stained chromosomes by structured illumination microscopy, which provides increased spatial resolution²³. Analysis of presented images provides independent support for the presence of multiple Zip1 signals that exhibit spacing predicted from the combination of focal and domainal triad/crossover sites described above (Supplementary Fig. 6E – G and Supplementary Note – Super-resolution images of the synaptonemal complex)".

Interestingly, however, analysis of this data suggests that, paradoxically, "higher resolution" analysis is less useful in defining two sets of signal, as described in our response to Reviewer 4.

3. We strongly disagree with the reviewer's comments about "over-analysis" and "over-interpretation". In our view, there is no basis for such criticism. We stand solidly by all of the conclusions and analyses in this work, irrespective of this reviewer's views.

Reviewer #4 (Remarks to the Author):

1) There was a range of chromosome sizes used and chromosomes were not identified in these spreads. The density of the chromosomes of the Zip3 foci at smaller sizes are different than those of the larger sizes. Given the way the periodicities were identified, it seems that the identified frequencies would be changed depending on the chromosomes used. Also edge effects would come into play given the low number of foci per chromosome. An experiment that would address this is to divide the spreads into small and large chromosomes and see if there is the same periodicities obtained.

There is a fundamental misunderstanding on the part of this reviewer. The density of Zip3 foci can be defined either per kb of DNA or per micron of axis/SC. The two metrics are different because the "packing ratio" of kb/micron axis is variable for different regions along chromosomes (for discussion see Zhang et al. (2014). PLoS Genetics; 10: e1004042, Fig. S6 and associated text). It is now known that the relevant metric for crossover interference (the subject of the current paper) is micron axis length (Zhang et al., (2014). Nature 511:551-556). And the number of Zip2 or Zip3 foci per micron axis length is the same for all analyzed chromosomes over a wide range of chromosome sizes, as we now document in Supplementary Fig. 4I.

Specifically, as shown in Supplementary Fig. 4I, data from our laboratory shows that chromosomes III, XV and IV, whose axis/SC lengths range four-fold (mean lengths of 1.2, 3.2 and 4.8 μm

respectively), have very similar densities of Zip3 foci per μm axis/SC (1.50, 1.47 and 1.42 foci/ μm respectively). And, in an independent study from another laboratory (Fung et al. (2004). *Cell*; 116: 795 – 802), the density of Zip2 foci along Chromosome XV was 1.46 foci/ μm (the same as our analysis, 1.47 foci/ μm). Thus, there is no evidence to suggest that “the identified frequencies would be changed depending on the chromosome used” or that edge effects (which will be more prominent on smaller chromosomes) are of a concern.

The reviewer is likely thinking of data that defines crossover density as foci per kb of DNA (Fung et al. (2004). *Cell*; 116: 795 - 802). This study shows that the density of foci per kb of DNA does vary among different chromosomes. We observe the same variation for Zip3 foci per kb of DNA in our laboratory's experiments. Similar conclusions regarding crossover frequencies per kb have been presented by other studies that measured crossovers by genetic or DNA analysis (e.g. Chen et al. (2008). *Dev Cell*; 15: 401 - 415). But these findings are not relevant to the current work (above).

We now discuss and clarify these complexities in detail in a Supplementary Note – Crossover density and chromosome length (line 176 – 220 of Supplementary Information).

The fact that variations in length do not affect our conclusions is further bolstered by additional considerations.

(a) The range of chromosome lengths for the analyzed chromosomes is extremely narrow ($3.15 \pm 0.4 \mu\text{m}$) and is essentially identical to that for chromosome XV alone, as we now show in Supplementary Figure 3A, top versus middle bottom. Thus, neither significant variations in length nor the absence of identification of specific chromosomes, another concern of the reviewer, are relevant. [That is, we would have had the same variation in length if we had specifically examined chromosome XV.]

(b) The above considerations also exclude any significant role of “end effects”. End effects will be more prominent on shorter chromosomes. But the density of Zip3 foci (as well as CoC relationships) are the same for (i) for chromosome XV which is the second largest yeast chromosome; and (ii) for chromosome III, which is one of the three smallest yeast chromosomes (Supplementary Fig. 4I; CoC analysis in Zhang et al., (2014). *Nature* 511:551-556).

(c) In accord with the reviewer's suggestion, we divided the measured chromosomes into the shortest and longest 50% and show that the distribution of distances between adjacent peaks (and thus their periodicities) is the same in both groups, for both focal and domainal peaks (Supplementary Fig. 4H). This is, of course, expected given the above considerations that the density of Zip2/3 foci is independent of chromosome length plus the very small variation in lengths among the examined chromosomes.

The Results section summarizes the above information and directs the reader to the relevant supplementary material as follows:

Line 105 - 111: “Chromosomes selected for tracing were narrowly distributed around $3.15 \pm 0.4 \mu\text{m}$ in length (mean \pm standard deviation, Supplementary Fig. 3A top), with a very similar chromosome length profile as chromosome XV, the second largest chromosome in yeast, comprising 1.1 Mb ($3.14 \pm 0.3 \mu\text{m}$; Supplementary Fig. 3A middle). Additional considerations show explicitly that variations in chromosome length and/or identity in the analyzed sample do not impact the conclusions below (Supplementary Note – Crossover density and chromosome length).”

Finally, in accord with the considerations above, we would like to note that any significant effect of length, or any end effect, would have the consequence of creating noise, thus tending to obscure any basic pattern(s). Instead, the striking finding of the current work is the discovery of significant, coherent and sensible patterns which match and explain many other findings.

2) Zip2 and Zip3 foci don't overlap exactly, but it is assumed that they do and thus can be used interchangeably. In the literature, Zip2 foci seem to be significantly fewer than Zip3 foci. It thus is not clear one can combine the data from both.

Our reading of the literature is different from that of the reviewer. We think that the reviewer is referring to the findings of Agarwal and Roeder (2000) where they observe “57 +/- 6 (20 nuclei scored) Zip3-staining foci per pachytene nucleus” and 66 +/- 8 Zip2 foci per nucleus (42 nuclei scored)”. The authors further state that “Zip3 almost completely colocalizes with Zip2 at the pachytene stage (95% +/- 4% of Zip3 foci contain Zip2, 99% +/- 2% of Zip2 foci contain Zip3, 19 nuclei scored).” In this context, our response to the reviewer's points is as follows:

1. 95% and 99% colocalization is dramatically high, much higher than in many other studies of presumptive crossover-correlated foci. Thus, we disagree with the reviewer statement that “Zip2 and Zip3 foci don't overlap exactly”. Furthermore, failure of perfect colocalization is easily explained by either/both of two considerations. (a) different numbers of foci of Zip2 and Zip3 because of differences in the way images were thresholded, differences in the levels of background staining, and/or (we can now surmise) occasional obfuscation of “focal” peaks by “domainal” peaks. (b) recent timelapse studies in *C. elegans* (S. Kohler, unpublished) show that some recombination foci “blink”, implying that the numbers of molecules in each focus is variable and thus that, depending on the detection threshold, not all foci are detected in every image. We also consistently see colocalization of Zip2 and Zip3 foci in our experimental protocol (Supplementary Fig. 4D). Thus, cytological data does not support any concern that Zip3 and Zip2 foci don't overlap exactly. We now present a discussion of this point in a Supplementary Note – Zip2 and Zip3 foci mark the sites of canonical crossovers (line 222 – 232 of Supplementary Information).

2. Other data support the view that Zip2 and Zip3 foci mark the same sites. We have previously shown that CoC curves for Zip2 and Zip3 foci along chromosome XV overlap with another (Zhang et al. (2014). *Nature*; 511: 551 – 556 (Fig. 5A). Importantly, this overlap is observed despite the fact that this data compared results from two independent sources: Zip3 measurements made in the Kleckner lab (Zhang et al., 2014) and Zip2 measurements made by the Roeder lab (Fung et al. 2004). Correspondingly, in these data sets, the distribution of distances between adjacent foci is the same for both Zip2 and Zip3 (Supplementary Fig. 4E).

3. In the current study we have analyzed and presented data for Zip2 and Zip3 independently, and thus, contrary to the reviewer's comment, do not ever combine data for both types of signals.

4. We get the same conclusions in our analyses for both Zip2 and Zip3, despite independent analyses, as we state in the paper.

Readers of the current paper are directed to the above information as follows:

Line 212 - 216: “Zip2 is a member of a three-protein complex that mediates the DNA/structure interface^{13, 14} whose cytologically-visualized foci exhibit the same spatial distribution as those of Zip3 foci¹⁵ (Supplementary Fig. 4D, E and Supplementary Note – Zip2 and Zip3 foci mark the sites of canonical crossovers) and thus also mark the sites of “canonical crossovers”^{5, 7}.”

In summary: the fact that Zip2 and Zip3 both mark the sites of crossovers is the paradigm in the field. We find no reason to doubt this conclusion and the corresponding results in the current study.

3a) The distribution of Zip1 and Hop1 are much less punctate than Zip2 and Zip3 foci, thus you would always pick up some periodicity in Zip1's and Hop1's signal that would match periodicities that would be found in Zip2/Zip3 as well as potentially other periodicities. Reviewer 2 has a point that some of the focal peaks in Hop1 and Zip1 are minor which is part of the worry that both Zip1 and Hop1 are distributed throughout the chromosomes and that one can always find periodicities that will match Zip1/Hop1.

If we understand correctly, the reviewer thinks that Zip1 and Hop1 signals could be only broad and domainal but with random irregular lumps, rather than the combination of patterned domainal and focal signals as our analysis suggests. If our understanding is correct, we do not find this possibility to be plausible. There is no reason that random lumps within a broad domainal signal should (a) occur with the same periodicities for both Hop1 and Zip1, much less (b) occur in triads that correspond to the sites of Zip3 focal signals, which in turn correspond to sites of previously-defined Zip3 foci.

We also note that focal signals for Zip1 and Hop1 do not reflect artifacts of spectral bleed-through, as we document extensively in Supplementary Figure 1.

An additional argument against the reviewer's point is provided by analysis of published SIM images of Zip1 which, in brief, resolves multiple peaks of intensity whose spacings correspond exactly to those identified and predicted by our analysis for the combined effects of focal and domainal triads (details below).

3b) An experiment that would make these results more convincing is to show that this same distribution of focal peaks does not occur using a protein that is not associated with crossovers but is also found along the axis.

There are two difficulties with this suggestion.

(a) Prior to this analysis, most people would have said that Hop1 and Zip1 were not associated with crossovers and are found along the axis.

(b) As far as we are aware, every axis component is associated with, and relevant to, crossing-over, because of the intrinsic nature of the meiotic process in which axis/recombination communication plays a central role. Thus, identifying a molecule that is specific to the axis, without any role in recombination, would seem to be a fishing expedition (or, perhaps, a future experiment).

4a) Reviewer 2 raises an important point that the authors did not use super-resolution microscopy which would better define actual focal positions. There is a strong possibility that the authors may get entirely different results if they could actually distinguish foci that are close together as distinct foci. At their resolution these foci would be merged and if one does not have an accurate depiction of the foci, it makes the analysis less convincing. An experiment that does not require reacquiring imaging data from the spreads is just to show that in the majority of cases that the distribution of foci using super-resolution is not significantly different than that obtained using conventional fluorescence microscopy for one identified chromosome. It would require marking a specific chromosome but that is relatively easy in yeast.

1. We would like to re-emphasize that resolution is not an issue for our identification of four prominent periodicities corresponding to two tiers of triads, the shortest of which is ~460 nm. We stated this previously in the text, but have now stated it there more explicitly: Line 143 - 145: "All identified periodicities are above the resolution limit of our imaging system (190 – 240 nm; Abbe's diffraction resolution limit of light)." Further, the ~460 nm distance is the same distance previously seen for canonical Zip2 and Zip3 foci, for which no issue has ever been raised concerning resolution.

2. We note that we also show that focal and domainal tend to be separated by a distance corresponding to the separation of recombination precursors, ~230 nm. This is just on the cusp of the resolution of our widefield imaging and image analysis. Our ability to define this separation distance reflects the fact that the data were separated into focal and domainal components, allowing the distances between adjacent peaks of the two types to be defined, essentially at single pixel resolution.

3. Super-resolution imaging data (e.g. from structured illumination microscopy (SIM)) cannot be acquired from the existing spreads; it would require an entirely new analysis on a different (marked) strain. Given the robustness of our results, this does not seem to use to be useful at this time. Furthermore, SIM analysis may actually be less useful than widefield imaging, for reasons discussed below.

4. A previous study has actually examined *Zip1* staining along yeast pachytene chromosomes by SIM analysis (Voelkel-Meiman et al. (2019). *PLoS Genetics*; 15: e100820). We have now analyzed quantitatively the *Zip1* intensity values along several of the chromosomes presented in published images. As we now show in Supplementary Fig. 6E-F and discuss in Supplementary Note - Super-resolution images of the synaptonemal complex (line 251 – 272 of Supplementary Information), the outcome of super-resolution images of the synaptonemal complex is an array of many peaks along each chromosome, with no obvious distinction between broad and narrow types. Furthermore, when we define the distances between adjacent peaks, we find that they fall into two discrete categories, of ~ 0.23 μm and ~ 0.46 μm . These are exactly the distances predicted for the combined effects of focal and domainal peaks, as shown quantitatively by the current analysis, with specific implications (Fig. 4E, F; Supplementary Fig. 6G; text lines 352 - 362). In brief: precursor recombination interactions occur every ~ 0.23 μm . Most focal peaks (corresponding to sites of previously-defined *Zip2/3* foci) are separated by ~ 0.46 μm , implying that they usually occur every other precursor, while domainal peaks occur at precursor sites that are interdigitated between sites of focal peaks and thus will usually be separated from a focal peak by ~ 0.23 μm .

5. Contrary to the above expectations of reviewers 2 and 4, SIM analysis may be less useful than our current widefield analysis. The straightforward interpretation of the SIM data analysis of *Zip1*, where no distinction between focal and domainal signals is immediately apparent, is that the methodology used to increase resolution has narrowed the broad signals that appear as domains in wide field imaging, thereby obscuring the difference between the focal and domainal signals. It is possible that detailed analysis of SIM images could tease out a meaningful result and thereby confirm/extend our conclusions. However, this is not obviously going to be the case. Thus, this exercise might provide further evidence for our conclusions, or it might, on the other hand, not be very useful in the straightforward way that Reviewers 2 and 4 envisioned. In either case, it is not required for the conclusions presented in the current work.

REVIEWER COMMENTS

Reviewer #5 (Remarks to the Author):

General comments. We appreciate this reviewer's efforts. Aside specific replies to reviewer comments below, we have made several improvements to the manuscript.

(i) We now include a Background section which provides context. Diverse comments from several reviewers make it clear that this is necessary, even for readers in the meiosis field. This includes relevant information about the relationship between Zip2 and Zip3 foci (see below).

(ii) We have modified the logic of the presentation, which is now more straightforward, and which makes the reasoning and implications more clear throughout.

(iii) We have added a short section at the end of the paper that points out the multiple convergences between FFT results and biological findings. This is important because these convergences provide strong support for the validity of all findings. For convenience, this section (lines 601-617) is copied below.

(iv) We have added critical evidence from the literature which provides strong evidence that the properties we define for longer periodicity triads, and link to previously-defined "minority crossovers", are applicable across diverse organisms (Discussion).

lines 601-617

"Convergence of mathematical and biological findings. The results presented above illustrate the power of integrating mathematical analysis of periodicities with functional and biological considerations. FFT analysis defines two prominent fluctuations in signal intensity, of shorter and longer periodicity, which are the same for all three molecules. iFFT analysis reveals that the fluctuations of the three molecules are in phase, with resultant clustering of component peaks in well-separated ~200 nm triads. Biological data directly link shorter periodicity triads to sites of canonical crossovers. Biological data confirms the validity and functional significance of longer periodicity triads for recombination, by four types of observations. First, longer periodicity triads are affected by mutations known to affect interference for canonical crossovers. Second, longer periodicity triads are differentially affected by absence of Pch2. Third, the distance between a longer periodicity triad and the nearest shorter periodicity triad is the same as the known distance between adjacent recombination precursor sites. It seems unlikely that this is a coincidence; instead, it implies that longer periodicity triads arise at sites of recombination events, along the same set of precursors as canonical crossovers, in a particular (interdigitated) way. Fourth, the number of longer periodicity triads and their effects on crossover interference relationships correspond directly to those observed for minority crossovers, supporting a new explanation for these events."

My specific comments regarding the issues raised by Refs 2 and 4, and of other issues in the manuscript are described below.

1. I am not sure I understand how the bleedthrough analysis in Supp Data 1 really proves there is no bleedthrough. This should have done in separate experiments where only ONE of the proteins is labeled and the three channels are acquired.

The previously presented analysis rigorously demonstrated absence of bleedthrough. We have retained this analysis because we feel that it is especially useful, but we now provide a clearer explanation of the approach. In addition, we have also added the more traditional approach that the reviewer requests.

The previous approach (now Supplementary Fig. 4A, B) is as follows. In any preparation, each of the three secondary antibodies forms non-chromosomal aggregates which comprise pure signals for each of the respective fluorophores. Since the aggregates of different secondary antibodies do not colocalize, these aggregates can be assessed for bleedthrough into other acquisition channels. No bleedthrough was detected for aggregates of any of the three fluorophores. This analysis has the unique advantage that it allows the assessment of bleedthrough in actual experimental samples that have been stained for all three molecules.

The approach the reviewer had in mind is now presented as Supplementary Fig. 4CD. There is no bleedthrough from any fluorophore in any of the two other acquisition channels.

2. While length analysis is consistent with their traces corresponding to chromosome XV, it is unclear why the authors did not label this chromosome directly, ie. by using oligopaints. As other chromosomes have similar lengths (VII, XII), this puts into question of whether they are really looking at chr XV. However, this should not be an issue if the frequencies of the spots are in um as argued by the authors.

The reviewer has misunderstood the sentences in the text. In our study, specific chromosomes were not identified. It just happens, by chance, that the length distribution of the measured chromosomes matches that of chromosome XV, which we had previously analyzed specifically in published work. This correspondence is not surprising given that, as the reviewer points out, several yeast chromosomes are of similar length. We have now reworded the relevant section to make the situation more clear (lines 173-178):

"Specific chromosomes were not identified. However, chromosomes selected for tracing were narrowly distributed in length around $3.15 \pm 0.4 \mu\text{m}$ (mean \pm standard deviation; Supplementary Fig. 3), very similarly to chromosome XV, the second largest chromosome in yeast, comprising 1.1 Mb ($3.14 \pm 0.3 \mu\text{m}$; Supplementary Fig. 3). Critically for our analysis, bleed-through of each signal into the acquisition channels for the other two signals is negligible (Supplementary Fig. 4)."

We also note that this information was included in the previous text in part because of a concern, raised by a previous reviewer, and touched upon by this reviewer, that variations in length among different traced chromosomes could potentially influence FFT results. We have addressed this issue in the context of our added FFT analysis which specifically considers the effect of confining the signal to the observed chromosome length of $\sim 3\mu\text{m}$ (point (5) below; Supplementary Fig. 6).

3. The nature of the signal (diffraction limited images) is such that it is difficult to visualize directly the structures the authors are aiming to detect. A convincing validation of whether their algorithm is able to deconvolve multiple overlapping signals is absent.

We now present additional FFT analyses which address this issue in detail (see comment (5) below).

4. On point 4.2, ref 4 argues that Zip2 and Zip3 should not be used interchangeably. The data (Supp Data 4D) provided by the authors is not convincing in supporting the authors' statement that these two factors can be interchangeably used, as there are evident changes in the spatial distributions shown. Therefore, while the distributions of these factors seem to be very similar, they are not identical.

We disagree with this concern.

First, the fact that these two molecules colocalize and mark the sites of canonical crossovers is dogma in the field. Studies of budding yeast meiotic recombination for the past 20 years have used wide field imaging of fluorescent foci of Zip2 and Zip3, specifically, to define canonical crossover recombination ensembles and to analyze crossover patterns along meiotic pachytene chromosomes. Correspondingly, Zip2 and Zip3 foci have long been known to colocalize (e.g. Agarwal and Roeder, 2000: Abstract: "Zip3 protein colocalizes with Zip2 protein at discrete foci along meiotic chromosomes"). Moreover, all available evidence suggests that the two molecules are part of the same ensemble of molecules, called "ZMMs", which are known to directly interact to specifically mediate early stages of crossover formation.

Second, in images that we presented, Zip2 and Zip3 were visualized in the same sample in the conditions of the current study (now Supplementary Fig. 1). The reviewer refers to "evident changes in the spatial distribution". We aren't sure exactly what the reviewer means by this phrase (i.e. "changes" versus "differences"?). However, he/she seems to think that the two signals should be identical. This is not the correct expectation. The two molecules are each present in multiple copies within very large (<200nm) ensembles. But they are not directly interacting (e.g. as heterodimers) and thus will be at different positions within these ensembles. And these large ensembles can have diverse orientations relative to the viewing plane and relative to the long axis of the chromosome, both of which will affect the shapes of the corresponding signals and the exact relative positions of their intensity centroids in these 2D images. Moreover, these complexes undergo dynamic conformational modulations during the period of meiosis when their positions are defined, occurring in couplets at earlier stages and singlets at later stages, in correspondence to changes in the status of the

two involved double-strand break ends. The expected variation in the states of such complexes and their components as visualized by SIM is illustrated for other components of canonical crossover recombination complexes along pachytene chromosomes of *C.elegans*, which show complexities of size, shape, internal organization, and orientation relative to the longitudinal axis of the chromosome (Scale bar 500nm). from: Woglar A, Villeneuve AM. Dynamic Architecture of DNA Repair Complexes and the Synaptonemal Complex at Sites of Meiotic Recombination. Cell. 2018. 173(7):1678-1691.

[REDACTED]

In the image samples presented in Supplementary Fig. 1, the peaks of maximal intensity for the two molecules within a given image differ by 1-2 pixels ($\leq 135\text{nm}$). This is well within the range expected for two sets of molecules within the same $\sim 200\text{nm}$ complex that is present in diverse orientations in different snapshots as described above.

Third, *in the present study*, the results observed for the two molecules are indistinguishable, as shown for every analyzed feature:

- Two-component Gaussian mixture models of the distribution of two most prominent FFT periodicities (Fig. 2C; Supplementary Fig. 5A).
- Distribution of distances between adjacent iFFT-defined peaks for both shorter and longer periodicities (Figs. 2H and I; Supplementary Figs. 5B and 5C).
- Evidence for the presence of triads, for both shorter and longer periodicities (Figs. 2J and 2K; Supplementary Figs. 5D and 5E).
- The nature of sub-component periodicities for the distributions of distances between adjacent iFFT-defined peaks for both shorter and longer periodicities (Figs. 3E and 4B; Supplementary Fig. 9)
- Crossover interference (CoC) relationships in wild type meiosis for both shorter and longer periodicities (Figs. 3C and 4A; Supplementary Figs. 7A and 10A).
- Distribution of distances between each longer periodicity peak and the nearest adjacent shorter periodicity peak (Fig. 4C; Supplementary Fig. 10B).
- The 1:2 ratio of the number of longer:shorter periodicity peaks (Fig. 5A; Supplementary Fig. 11A).
- CoC relationships for total (shorter + longer) periodicity peaks (Fig. 5C; Supplementary Fig. 11B).

5. I am not convinced of the Fourier analysis used. Typically in such analysis one expects to find directly peaks in the Fourier space, which they don't seem to find (e.g. Supp Data 3B). Instead they seem to rely on derivative analysis to identify

changes in the slope of the Fourier function. I have never seen this analysis in the past and they do not show solid evidence of why it has to be used or that it works.

We understand why the Reviewer might be concerned. We now present the summed data in position space, rather than Fourier space (details in Methods, line 891). Moreover, further analysis shows that, for each of the three molecules, the sum of all periodicities can be divided (deconvolved) into two groups of heterogeneous shorter and longer periodicities, of $\sim 0.5\mu\text{m}$ and $\sim 1\mu\text{m}$, in direct correspondence to the two major groups seen in the distribution of the two-most-prominent peaks (Fig. 2C). The results of such analyses are now presented in the text (lines 220-261 and Figs. 2D, E), with additional information and controls in Supplementary Fig. 6.

A brief summary of all FFT analysis now presented is as follows.

(i) FFT analyses of experimental intensity profiles for each molecule along each chromosome were performed (Fig. 2A, B).

(ii) Since visual inspection suggests that there are two types of signals, narrow and broad, for all three molecules (Fig. 1D-F), we first defined the two most prominent FFT peaks, in Fourier space, in each trace of each analyzed chromosome, and presented the distribution of the corresponding periodicities for each molecule. We have now presented two-component Gaussian mixture models of these distributions, in accord with the possibility of two categories of signals (Fig. 2C). These fits show the presence of two groups of signals of shorter and longer periodicity ($\sim 0.5\mu\text{m}$ and $\sim 1\mu\text{m}$, respectively, for all three molecules). [(note: We previously presented four-component Gaussian mixture model fits to these data. These four distributions reflect the fact that the shorter and longer periodicity signals are each made up of two subgroups. The existence and basis of these periodicities is clear from other information, but it is a secondary point, and it is now discussed later in the paper.)]

(iii) FFT outputs for all experimental signal intensity profiles (of a given molecule) are summed in k -space and then converted to position space (details in Methods, line 891). Hop1 and Zip1 give two peaks centered at greater and lesser than $\sim 0.7\mu\text{m}$ (Fig. 2D; reproduced below). For Zip3, the signal is not so simple because the shorter periodicity signal is much more prominent for this molecule (text; Fig. 2D and below).

(iv) Further analysis shows that the sums of FFT outputs are made up of two groups of heterogeneous shorter and longer periodicities of $\sim 0.5\mu\text{m}$ and $\sim 1\mu\text{m}$. The logic of the approach is as follows. The existence and nature of the shorter periodicity signal is not in doubt. We present extensive evidence in the text that this periodicity signal corresponds in every possible way to the periodicity known for canonical crossovers as defined by Zip3 and Zip2 foci by standard analyses ($\sim 0.5\mu\text{m}$). We reasoned that we could (a) simulate the FFT signals predicted for this shorter periodicity, including heterogeneity in height and spacing; (b) determine the sum of many such signals exactly as for the experimental data; and (c) subtract the resulting sum from the total experimental data peak to reveal the presence (or not) of longer periodicity signals. The outcome of this analysis, for each of the three molecules, is that the residuum after subtraction is a broad (heterogeneous) peak centered at $\sim 1\mu\text{m}$ (Fig. 2E and below). Furthermore, the longer periodicity signals are lower for Zip3 than for the other two molecules, in accord with lesser prominence of the longer periodicity

signal known from Zip3 cytological images and also visible in the distributions of two-most-prominent periodicities. These results show that the distribution of total FFT signals for the experimental data is composed of two groups of shorter and longer periodicities, of $\sim 0.5\mu\text{m}$ and $\sim 1\mu\text{m}$, both of which are heterogeneous.

(v) This analysis included controls which show that the algorithm used is accurate; that there can be artifactual longer periodicity signals that result from confining the shorter periodicity signal to $\sim 3\mu\text{m}$; but that the experimental data contains real, longer-periodicity signals.

(D, E) The sums of Fourier transform periodicities for all chromosomes can be deconvolved into shorter and longer periodicity groups centered at $\sim 0.5\mu\text{m}$ and $\sim 1\mu\text{m}$. (D) For each molecule, all individual Fourier transforms were summed in k-vector space and the sum converted to a distribution of probabilities in position space (μm ; Methods). (E) Solid grey line: the distribution of total periodicity probabilities was determined for simulated chromosome data that matches the shorter periodicity ($\sim 0.5\mu\text{m}$) peaks visible in primary experimental data and known for Zip3 foci (Supplementary Fig. 6; Fig. 3D). When this distribution is subtracted from the experimental distribution of total periodicities (D), the residuum (dashed grey line) exhibits no periodicities at $\sim 0.5\mu\text{m}$ (blue arrow). Thus, shorter periodicities in the data are fully explained by the subtracted shorter periodicities. A significant residuum of longer periodicities occurs

(dashed grey line) centered at $\sim 1 \mu\text{m}$ (red arrow), over and above longer periodicities present in simulated data (gold arrows), implying that this set of longer periodicities is present in experimental data (see also Supplementary Fig. 6).

6. The authors argue that spatial resolution is not an issue. However, I would side with the reviewers' arguments as the resolution of the images shown (e.g. Fig. 1) shows ill defined spots, with intermediate densities within spots that could actually arise from additional clusters not resolved by conventional microscopies.

While SIM doubles the resolution of these microscopies, the increase in resolution is often not sufficient. I would highly recommend the authors to use either STED or STORM, as these would afford a 5x or 10x increase in resolution that would settle the issue.

I don't agree with the authors' argument that SIM or other methods cannot be used with their existing samples as SIM/STED/STORM could be applied using immunofluorescence as the authors use for conventional imaging.

We continue to have a different point of view than the reviewers, for the following reasons:

A. Our analysis defines functionally important periodicities of two types of signals. The existence of these signals is not dependent on actually "seeing" the entities to which they correspond in primary cytological images. However, one can easily envision the two types of signals as follows. Both types of signals are nucleated at specific positions. The peaks of the triad signals that define the two periodicities are, in both cases, clustered within a short distance ($<200\text{nm}$) which defines the nucleation points. The difference in the two types of signals relates to how the molecules spread outward from those nucleation sites. The nature and complexity of shorter periodicity signals, i.e. of sites of canonical crossover complexes, is discussed above (point (4)). In essence, those signals are confined to a defined region, without extensive spreading. In contrast, for the broader spreading, longer periodicity signals, nucleation is similar, but the signals spread more widely along the chromosomes. There is no evidence for complexity in these signals (see below). Of course, it would not be shocking if they did not exhibit a perfectly smooth gradient at the level of individual molecules, given the complexity of the spreading ensembles. Regardless, any heterogeneity in the spreading portion of either type of signal would not in any way alter the reality of the existence of the two types of periodic signals.

B. There is no feature visible in cytological images that leads us to suspect the presence of any significant "undetected major periodicities". In fact, the newly added FFT analyses ((5) above) provides evidence that the two major types of signals account for all significant periodicities. Moreover, iFFT analysis shows that the triads for the two types of signals do not occur at the same positions but, in fact, are usually separated by exactly the distance between adjacent recombination precursor positions as known from previous analyses (see also Point (7) below). Thus, there are (only) two distinct periodicities.

We are not sure what are the "ill defined spots" to which the reviewer refers. But we guess that they correspond in position to the "focal peaks" visible in the intensity traces which, our analysis shows, correspond to the shorter periodicity/canonical

crossover signals. These signals are especially poorly defined for Zip1, and not so well defined for Hop1, and, in all cases, overlap with longer periodicity signals, and thus might appear as "ill-defined spots with intermediate densities". We have now added a new panel, Fig. 1D, to make the correspondence between visual features and the two types of signals more clear. In summary: if we are correct in our understanding of the reviewer comment, the feature they think are "unresolved signals" are, in fact, the specific signals that our analysis now identifies as narrow, shorter periodicity signals superimposed on the broader, longer periodicity signals with which they overlap.

C. The reviewer's comments suggest that they think there will be "structures" that correspond to the two periodic features (3 above). This may not be the best way to think about the identified signals, as discussed in point (A) above. We now make this more clear in cartoon form in text Fig. 4E.

D. We accept that higher resolution always has the potential to produce more information. However, we think that the current situation is a case in which lower resolution is actually more powerful than higher resolution. For example: if you look at a forest from the ground, you will see lots of trees of different types in various local relationships; but if you look from the sky, you can see broad groups of trees of different types with intermingled and complex borders. Inferring the latter pattern from mapping the positions of individual trees might be possible but is not the most efficient or appropriate approach. Put another way: by high resolution analysis, you "can't see the forest for the trees" while, by lower resolution analysis, the patterns of the forest are clear.

E. To carry out any high resolution study is a major endeavor with many technical challenges. Some relevant issues are as follows.

- Shorter and longer periodicity signals overlap. Thus, regardless of what type of imaging is used, the definition the two features will require analysis of longitudinal intensity fluctuations, and application of periodicity analysis, for all three signals, which is challenging (see below). There may not even be two features that can be seen as individual entities directly, especially for Zip1 and Hop1.

- We cannot simply take existing samples and image them in another system, for several reasons. Among these is the fact that different parts of the relevant molecules are located in different exact positions within the ensembles of interest. For example, N- and C-termini of Zip1 are located in the middle, and on the edges of the SC, respectively. Higher resolution analysis requires creation of additional tagged species with tags at known positions, a nontrivial endeavor.

- At higher resolution, the issue of "lateral" versus "longitudinal" localization emerges. It is possible, for example, that focal signals occur between axes while domainal signals occur along axes. But even more complex patterns are equally possible. Defining distributions of single molecules, or intensities of SIM objects, in this situation is not trivial. Sorting out these complexities, which are of course interesting, is a major endeavor.

- The complexity of single molecule analysis, and the challenges involved in identifying periodicities of molecules along the chromosomes, are illustrated by single molecule 3D STORM images of the Hop1 analogue HIM3 along a short segment of C.elegans chromosomes (examples below; from Kohler et al., 2017). The distribution of

"trees" is complicated, and identification of "forests" would require a lot of statistical analysis.

- SIM analysis requires significant data processing to "sharpen" the signals. It is not clear that the outputs from such processing will faithfully reproduce the domainal "shape" that the reviewer wishes to see. Instead, it may create some artifactual shape, as we know can occur from our own experience in other work. Thus, this method is not suitable for the type of signals we envision.

F. Finally, for these and other reasons, any such study is well beyond the scope of the current work.

Colors in 3D-STORM images of HIM3 indicate localization along the optical axis, with red being closest to the coverslip and violet at the most distant position, as represented by the colored scale bar.

[REDACTED]

Köhler S, Wojcik M, Xu K, Dernburg AF. Superresolution microscopy reveals the three-dimensional organization of meiotic chromosome axes in intact *Caenorhabditis elegans* tissue. Proc Natl Acad Sci U S A. 2017; 114(24):E4734-E4743.

7. I am even confused by the rebuttal of point 4a, as the authors talk about periodicities of 0.23 μm , while in previous paragraphs they argued their conventional microscope is able to detect the spatial variations in the sample. However, 0.23 μm is at the resolution limit of the microscopy methods they used.

The reviewer has perhaps not understood the nature of the analysis. Probably we were not clear in our explanation. We never visualize 0.23 μm directly. Previously published data from the laboratory, based on other types of analysis, shows that the precursor complexes that give rise to crossovers (and thus the two types of periodic signals) are evenly spaced at $\sim 0.23\mu\text{m}$ (see Background section). In the present analysis, distances between different types of signals are defined from the outputs of iFFT. Specifically, we defined the positions of intensity peaks from iFFT outputs for the longer and shorter periodicity signals at a resolution of one pixel (0.067 μm). We then defined the distances between different types of peaks of interest. The accuracy of these positions rests on the accuracy of the iFFT output, not on resolution in the primary images. And the accuracy of inter-peak distances as defined in this way is validated by the very close correspondence between the inter-peak distributions and interference patterns for shorter periodicity peaks versus those for canonical crossovers as defined by standard methods (Figs. 3D, E).

More specifically: we used the distances between iFFT peaks in several ways. Notably, the distance between a peak of one molecule and the nearest peak of a different molecule is used to define the existence of triads. These distances, for all three pairs of molecules, are $\sim 0.07\mu\text{m}$, i.e. basically at the resolution of peak definition (Figs.

2J and K; Supplementary Figs. 5D and E). For shorter periodicity signals this is fully compatible with the known dimensions of canonical crossover complexes (above). Additionally, we defined the distance between each longer periodicity iFFT peak and the nearest adjacent shorter periodicity iFFT peak of the same molecule. The modal value of this distance is 0.27 μm (Fig. 3C and Supplementary Fig. 10B). This result shows: (a) that the two types of signals do not occur at the same position and (b) this distance corresponds to the distance between adjacent precursor sites (above) which shows that both types of peaks occur at recombination precursor sites and that a longer periodicity iFFT peak is usually immediately adjacent to a shorter periodicity iFFT peak (that is, that a minority crossover is usually at a precursor site that is immediately adjacent to a canonical crossover). This is a very striking and highly significant finding which is unlikely to be a coincidence.

8. Finally, it is hard to assess the responses to the issues raised by Ref 2, as I have no access to previous interactions with the original issues raised by Ref 2. This said, I agree that there are issues regarding reliability of their Fourier method to identify peaks, of whether there is bleed through, whether Zip2/3 can be used interchangeably, and finally of whether the spatial resolution is enough to properly support their interpretation and model.

All of these issues are fully addressed above.

REVIEWERS' COMMENTS

Reviewer #5 (Remarks to the Author):

The authors have provided plausible explanations to most of my concerns.

We appreciate the Reviewer's favorable reconsideration of our manuscript.

White et al. describe an analysis of immunolabeled chromosome spreads of yeast nuclei arrested in the *ndt80Δ* arrest.

They trace the immuno-labeling along chromosome midlines, and find that the observed patterns fit a composite periodic function, consisting of three components. The individual functions are interpreted as corresponding to fundamental chromosomal parameters, such as classical and 'minority' CO interference and **XXX???** (DSB interference?)

I 96-100

While it is appreciated that the authors carefully explain their word usage, I urge them to refrain from redefining "foci", even just for that manuscript. A focus is intuitively thought of as a dot-like structure that appears naturally segmentable from its surrounding. This is categorically different from an "intensity peak", which does not require to be segmentable. I believe the readers may be unnecessarily confused by having to cope with the "unusual" usage of the word "focus", it distracts from understanding the main message and it might leave readers long term confused. I suggest to stay closer to the definition and call them "intensity peaks" or "local maxima".

Somewhat connected Fig. 1B and C illustrates the issue. While a peak finding algorithm may identify 5 local maxima along the intensity trace, a cohort of 10 students will not see 5 foci. As for the showcased local maxima, it seems that for calling peak # 2 of Zip3, # 3 of Hop1 and # 4 of Zip1 there is no, or an extremely low local threshold required. They would not be called by eye and I am not entirely convinced that one should place weight on a local maximum without any 'quality margin' that safely distinguishes it from noise or artifacts.

Also, the example spread features a bright structure, known as the poly complex, which accumulates various excess meiotic proteins, especially Zip1 and Zip3. Depending on its size and its intensity, it can influence the intensity of chromosomal signals and form an artificial gradient at a distance. It is worth to state how the authors avoid or mitigate that obvious problem. The chosen picture does not show a strong gradient, but again, if the authors do not experience such gradients, this should be documented.

I106

The intensity signal along 81 individual chromosomes is a small sample set, considering the issues with significance that arise.

The "reverse FFT": It needs to be better explained, which information is idealized and thus lost, when looking at the reverse FFT.

Of course, clearly the reverse FFT allows to inspect the two power spectra separately - but, it is not clear, whether all the noise of

the measurements is properly represented in those strikingly clear pictures...